# PSBench: Editing Image via GUI Agents in Photoshop

Yinuo Zhang [1]   Zian Cheng [1]   Ziya Zhao [1]   Zongyu Li [1]   Bingshuo Liu [1]   Qingbin Liu [2]   Junxian Cai [2]   Xi Chen [2]   Zhiying Tu [1]   Dianhui Chu [1]   Xiaoyan Yu [3]   Dianbo Sui [1]

## Abstract

Photoshop is a professional image editing software whose complex multi-level menus, fine-grained operations, and layer-based non-destructive editing pose substantial challenges for automated agents. Existing GUI benchmarks and methods primarily target web interfaces and short-horizon, low-complexity tasks, falling short in modeling the multi-step decision-making and semantic understanding required by professional graphic software. We introduce PSBench, the first benchmark specifically designed for image editing in Adobe Photoshop, consisting of 600 human-annotated tasks across three difficulty levels, with tasks drawn from official tutorials and popular real-world workflows. PSBench covers core functionalities such as canvas adjustment, layer manipulation, and filter application, and provides fine-grained evaluation metrics tailored to each task category. Our experiments show that even the state-of-the-art system, Agent S3, achieves a success rate of only 18.09% on difficult tasks, indicating that GUI agents still face considerable challenges in operating complex professional software. Furthermore, human-in-the-loop evaluations reveal that MLLMs, when serving as interactive assistants, can significantly improve novice users' task completion rates and reduce operation time.

 https://github.com/zyn1216/PSBench

## 1. Introduction

Through simulating human interactions with graphical interfaces, Graphical User Interface (GUI) agents (Nguyen et al., 2024; Wang et al., 2025a) can automatically execute complex tasks and make intelligent decisions, thereby significantly enhancing software testing automation, improving user assistance, and driving the automation and intelligence of diverse workflows. These capabilities demonstrate great potential in improving efficiency, reducing human errors, and supporting the execution of complex multi-step tasks (Gur et al., 2024; Furuta et al., 2024).

In this paper, we explore the possibilities and prospects of applying GUI agents to the field of image editing. The key motivation stems from three core aspects: ❶ Despite the remarkable progress of diffusion-based image editing methods (Shuai et al., 2024; Huang et al., 2025), they remain deficient in aspects such as high-resolution fidelity, intricate lighting and shadow modeling, and background preservation (some cases are shown in Table 9). Yet, in day-to-day industry practice, skilled photo retouchers effortlessly deliver all these inside Adobe Photoshop via nondestructive editing. Nondestructive editing, the philosophy of Adobe Photoshop, refers to any workflow that allows an image to be modified—whether through adjustments, retouching, or compositing—without permanently altering the original pixel data, thereby preserving the ability to revisit, revise, or remove every edit at any future point. ❷ However, Photoshop is notoriously tricky for beginners, since its powerful but intricate interface demands substantial training and domain expertise to navigate effectively. ❸ Naturally, we expect GUI agents to lower the entry barrier for non-expert users engaging in image editing within Photoshop, yet current solutions remain far from satisfying practical demands. In detail, existing mainstream GUI agent benchmarks (Xie et al., 2023; Ma et al., 2024a; Furuta et al., 2024; Pahuja et al., 2025) primarily focus on web environments or general-purpose operating systems, like Webshop (Yao et al., 2023), OSWorld (Xie et al., 2024) and WebArena (Zhou et al., 2024), where tasks are relatively simple and lack domain expertise. Besides, these GUI agents often target everyday software accessible to non-experts such as Chrome or Word, with limited and uniform interaction modes that fail to capture the operational complexity of professional software, like Photoshop.

Therefore, we propose PSBench, a novel GUI agent benchmark focused on image editing tasks in Adobe Photoshop. Such a benchmark presents unique challenges for GUI automation: The interface of Photoshop is not only highly hierarchical and feature-rich but also depends on fine-grained,

---

[1]Harbin Institute of Technology [2]Tencent [3]Nanyang Technological University. Correspondence to: Dianbo Sui <suidianbo@hit.edu.cn>.

*Proceedings of the $43^{rd}$ International Conference on Machine Learning*, Seoul, South Korea. PMLR 306, 2026. Copyright 2026 by the author(s).

multi-step operations. For instance, Photoshop's core layer system requires agents to understand and manipulate non-destructive editing workflows, including layer order, masks, and adjustment layers. Moreover, many tools (e.g., brush, lasso, path) are context-dependent and parameter-sensitive, producing entirely different effects under different modes or environments. Furthermore, numerous operations involve pixel-level precision and parameter adjustments, demanding a level of accuracy far beyond that required by everyday software accessible to non-experts.

In task design of PSBench, we introduce three difficulty levels: easy and medium tasks are manually designed based on basic operations (e.g., cropping, flipping), while hard tasks are sourced from popular YouTube tutorials to ensure both realism and diversity. Ultimately, we construct a high-quality human-annotated benchmark comprising 600 tasks and more than 300 fine-grained evaluation functions, covering a wide range of key Photoshop functionalities such as layers, canvas, and filters, and reflecting diverse real-world use cases. For evaluation in PSBench, in addition to conventional task success rate metrics, we further propose the Non-Destructive Editing Consistency (NDEC) metric, designed to assess whether agents adhere to Photoshop's non-destructive editing philosophy. In such a metric, based on Adobe's official definition of non-destructive editing, we design a checklist (Ribeiro et al., 2020) including six core questions to compare reference operation trajectory provided by expert annotators and agent trajectory on a per-task basis, thereby enhancing the professionalism and granularity of benchmark evaluation.

Comprehensive evaluations on **PSBench** reveal that even state-of-the-art models struggle to reliably translate their visual-language capabilities into executable Photoshop operations, with overall success rates ranging from single digits to low teens. Specifically, the best-performing general-purpose MLLM, GPT-4o, achieves a success rate of only $17.46\%$ on non-layer tasks and drops to a meager $3.80\%$ on layer-intensive tasks. While hierarchical agentic frameworks like Agent S3 improve performance by combining high-level planning with UI execution, they still achieve only $18.09\%$ success on hard tasks, highlighting the formidable challenges posed by professional software.

However, beneath these low success rates, a deeper analysis using our NDEC metric reveals a significant latent strength: mainstream MLLMs demonstrate remarkable professional awareness, with NDEC scores exceeding $70\%$ and reaching as high as $95.83\%$ on simpler tasks. Their generated action sequences are often complete and professional, strictly adhering to non-destructive editing workflows. Leveraging this latent planning capability, we conducted human-in-the-loop experiments and found that these MLLMs can serve as highly effective *Photoshop Assistants*. In this collaborative

mode, the assistant provides step-by-step guidance while the human handles execution, leading to a $100\%$ success rate on easy tasks and a $75\%$ success rate on hard tasks—surpassing both autonomous agents and unassisted novices while reducing completion time by over $50\%$ compared to users with internet access. Consequently, for feature-rich and expert-level applications like Photoshop, shifting the paradigm from fully autonomous GUI agents toward a human-AI collaborative mode—one that pairs the model's deep semantic understanding with the human's precise operational execution—represents a more practical and high-impact research direction.

**Conflict of Interest Disclosure.** The authors declare that they have no financial conflicts of interest related to this work.

## 2. Related Work

**GUI Agent.** Currently, GUI agent development primarily follows three mainstream paradigms: The first category consists of general-purpose models, which possess broad capabilities, with "computer usage" being just one of many abilities that can be elicited through prompting. These models retain the capacity to perform other tasks such as dialogue and code generation, with typical examples including GPT (OpenAI, 2024; 2025a), Gemini (Comanici et al., 2025), Claude (Anthropic, 2024), and Qwen (Bai et al., 2025) series. The second category comprises specialized models, which are specifically trained for computer use agent tasks and lack the ability to perform other functions. Examples include AutoGLM-OS-9B (Lai et al., 2025), OpenCUA-32B (Wang et al., 2025b), and UITARS-1.5-7B (Seed, 2025). The third category involves agent frameworks, which integrate one or more general-purpose models with specialized models into structured workflows. These typically employ GPT-series models as planners, supplemented by dedicated or task-specific models as execution foundations, such as CoACT-1 (Song et al., 2026). Evaluations on the current authoritative benchmark OSWorld reveal a clear performance trend: agent frameworks > specialized models > general-purpose models.

**GUI Agent Evaluation.** Currently, benchmark evaluations for GUI agents can be broadly categorized into two main types: skill-specific evaluation and end-to-end task completion evaluation.

- **Skill-specific evaluation:** This type of benchmark is designed to assess a GUI agent performance in particular capabilities. The core competencies can be summarized into three aspects: (1) visual grounding ability, (2) reasoning and planning ability, and (3) action execution ability. Among these, the first two are especially critical, as they directly determine the agent's perceptual and decision-making capabilities in graphical interfaces. (Nguyen et al., 2024) In the field of visual grounding capability evaluation,

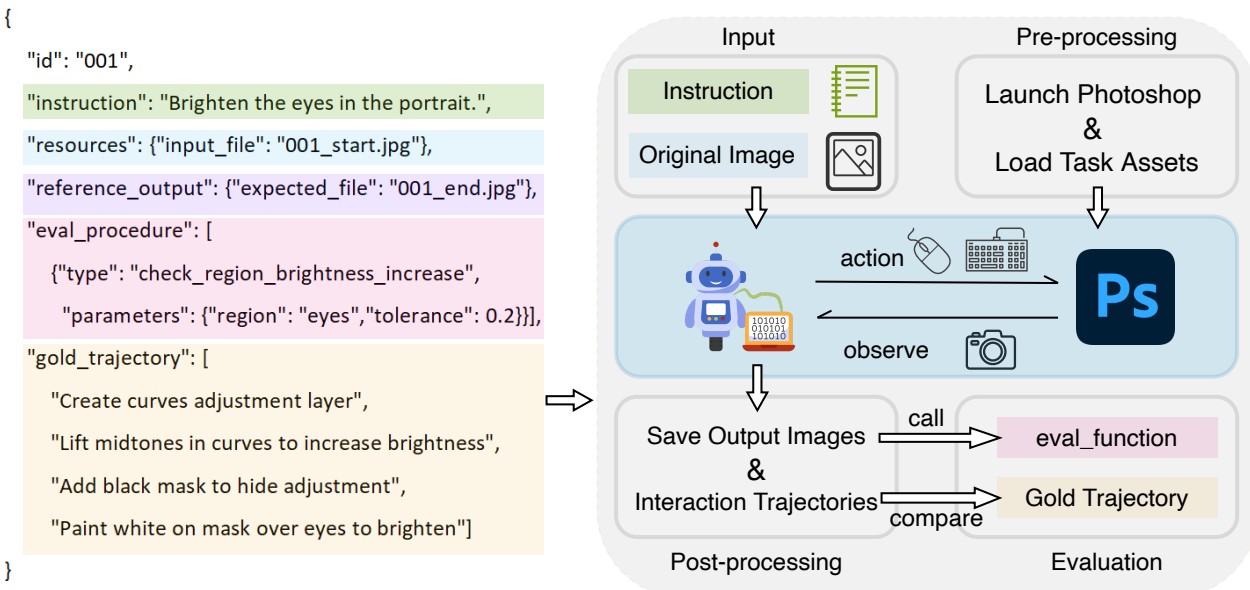

```
{
    "id": "001",
    "instruction": "Brighten the eyes in the portrait.",
    "resources": {"input_file": "001_start.jpg"},
    "reference_output": {"expected_file": "001_end.jpg"},
    "eval_procedure": [
        {"type": "check_region_brightness_increase",
            "parameters": {"region": "eyes","tolerance": 0.2}}],
    "gold_trajectory": [
        "Create curves adjustment layer",
        "Lift midtones in curves to increase brightness",
        "Add black mask to hide adjustment",
        "Paint white on mask over eyes to brighten"]
}
```

*Figure 1.* The overall framework of PSBench. The left part illustrates the task configuration: for each task, PSBench provides an instruction, input image resources, corresponding expected output images, and a gold trajectory. The right part demonstrates the actual interaction process of a GUI agent in the Photoshop environment: the GUI agent performs tasks by interacting with the environment through mouse and keyboard operations; the post-processing module saves output images and records interaction trajectories; the evaluation module invokes task-specific evaluation functions and compares the agent's trajectory with the gold trajectory.

a series of benchmarks have emerged: ScreenSpot (Cheng et al., 2024) and its improved version ScreenSpot-Pro (Li et al., 2025) support cross-platform UI localization and continue to advance in terms of realism and annotation quality. UI-I2E-Bench (Liu et al., 2025) and UI-Vision (Nayak et al., 2025) further extend this direction by aligning natural language instructions with GUI elements of varying scales and types, thereby enhancing the generalization ability of language-interface interaction. For reasoning and planning evaluation, offline benchmarks (Chen et al., 2025b; Li et al., 2024; Kapoor et al., 2024) primarily assess a model's ability to predict actions based on fixed interaction trajectories, while online benchmarks (Bonatti et al., 2024; Rawles et al., 2025; Xu et al., 2024; Liu et al., 2024) enable interactive evaluation across platforms, placing greater emphasis on the agent's real-time reasoning and decision-making performance in dynamic environments.

- **End-to-end task completion evaluation:** These benchmarks place GUI Agents in interactive environments such as Android emulators, virtual machines, or web-based setups, and require them to accomplish holistic tasks from start to finish. Representative efforts include those targeting mobile devices (MobileAgentBench (Wang et al., 2024), SPAbench (Chen et al., 2025a), AndroidLab (Xu et al., 2024)) as well as those designed for web and desktop applications (OSWorld (Xie et al., 2024), WebArena (Zhou et al., 2024), WebCanvas (Pan et al., 2024), Windows Agent Arena (Bonatti et al., 2024), WorkArena (Drouin et al., 2024)).

However, existing benchmarks generally lack dedicated evaluation for professional design software such as Photoshop. Most focus only on general-purpose software like Word or Chrome. Even in benchmarks that include tools like GIMP, e.g., OSWorld (Xie et al., 2024), the included tasks remain relatively simple (see Table 5 in Appendix for specific cases). Given the significant differences in interaction logic, task complexity, and operational granularity inherent to professional software, there is a clear and pressing need to develop a benchmark tailored to the characteristics of complex professional applications, with task designs that better reflect real-world usage scenarios.

## 3. PSBench Environment

PSBench operates on a locally installed Adobe Photoshop as the interactive environment. In PSBench, each task can be modeled as a partially observable Markov decision process (POMDP) $(\mathcal{S}, \mathcal{O}, \mathcal{A}, \mathcal{T}, \mathcal{R})$. Here, $\mathcal{S}$ denotes the state space, $\mathcal{O}$ denotes the observation space (see §2.1), $\mathcal{A}$ denotes the action space (see §2.2), $\mathcal{T} : \mathcal{S} \times \mathcal{A} \rightarrow \mathcal{S}$ denotes the state transition function, and $\mathcal{R} : \mathcal{S} \times \mathcal{A} \rightarrow \mathcal{R}$ denotes the reward function.

At each interaction step, the agent generates an executable action $a_t \in \mathcal{A}$ based on the current observation $o_t \in \mathcal{O}$. The action is executed in the environment to produce a new state $s_{t+1} \in \mathcal{S}$ and a new partial observation $o_{t+1} \in \mathcal{O}$ (e.g., the updated screen screenshot). The state transition function $\mathcal{T}$ determines the dynamics of the environment,

*Table 1.* Some mouse and keyboard actions in PSBench.

| Action Name | Description |
| --- | --- |
| WAIT | Pause operations for interface response |
| FAIL | Declare task failure and terminate |
| DONE | Declare task completion and end |
| click(x, y) | Click at specified coordinates (x, y) |
| dragTo(x, y) | Drag from current to target position (x, y) |
| write('text') | Input text content in current field |
| press('b') | Select brush tool in Photoshop |

while the reward function $\mathcal{R}$ provides immediate feedback depending on the task completion status. This interaction loop continues until the agent triggers a terminal signal (DONE or FAIL, see §2.2) or reaches the maximum step limit.

### 3.1. Observation Space

The observation space $\mathcal{O}$ in PSBench is designed to closely reflect the complexity of real human–computer interaction, and is defined as the union of text and image modalities:

$$\mathcal{O} = \mathcal{O}_{\text{Text}} \cup \mathcal{O}_{\text{Image}}. \tag{1}$$

The image modality consists of full desktop screenshots of the Photoshop workspace, including key UI elements such as the toolbar, layer panel, properties panel, and menu bar, as well as mouse position and cursor shape (e.g., precision cursor during selection). The screenshots also capture task-relevant canvas content, such as layer order changes, filter previews, and selection outlines, which reflect the real-time state and contextual dependencies of Photoshop operations. Compared to general applications, Photoshop exhibits a denser and more dynamic interface with highly modular functionality, requiring agents to perform precise UI element localization and stronger semantic understanding in order to operate effectively in such a complex and frequently changing design environment.

### 3.2. Action Space

The action space $\mathcal{A}$ in PSBench encompasses the full spectrum of human–computer interaction operations in Photoshop. Some action examples are shown in Table 1, including mouse movements, left/right clicks, multiple clicks, drag-and-drop operations, precise region selections, numerical inputs, and composite keyboard shortcuts (e.g., Ctrl+Alt+I to open the image size dialog). These actions drive Photoshop's core functional modules, such as menu commands, layer manipulations, tool switching, and canvas editing. Following OSWorld (Xie et al., 2024), we further introduce three special actions: WAIT (to wait for interface loading or filter rendering), FAIL (to declare task failure and terminate early), and DONE (to declare task completion and submit results). Action execution is implemented using the general-purpose Python library

pyautogui, enabling accurate reproduction of complex Photoshop interactions such as dragging to reorder layers, drawing paths, or entering color parameters. This design ensures cross-platform consistency and requires the agent to output syntactically correct and executable pyautogui code in order to accomplish specified tasks in Photoshop's dense, multi-state UI environment.

## 4. PSBench Benchmark

### 4.1. Data Collection

PSBench comprises a total of 600 diverse image editing tasks, collected and organized by four annotators proficient in Photoshop. Across the entire data collection process, four Photoshop-savvy annotators devoted approximately **270 working hours in total**. The detailed human effort could be found in Figure 2. The task construction process includes three main aspects:

**Task Collection.** Existing benchmarks, such as ASSIST-GUI (Gao et al., 2024), OSWorld (Xie et al., 2024), mainly focus on relatively simple operations, which fail to capture the complexity of real-world editing requirements. Unlike existing benchmarks, PSBench first organizes Photoshop functionalities into six operation-level categories: Geometric Transformations, Color and Tone Adjustments, Filter Effects, Selection Operations, Layer Operations, and Painting and Retouching tools. These categories serve as the basis for defining task complexity. Detailed category definitions and the complete list of fine-grained operations are provided in Appendix A.4.2.

PSBench categorizes tasks into three levels of complexity to enable multi-level evaluation:

- **Easy**: Tasks involving only a single category of operations.

- **Medium**: Tasks combining operations from 2–3 different categories .

- **Hard**: Tasks involving operations from more than 3 categories, corresponding to complex, real-world editing workflows.

Easy and medium tasks are manually created by annotators, who carefully examine the official Photoshop tutorials[1] to identify the most common basic operations and then manually formulated the corresponding task instructions. Hard tasks are derived from popular YouTube Photoshop tutorials[2], which cover topics like photo manipulation, photo effects, color effects, blend & retouching, text effects and much more. Annotators transcribe the high-level natural language instructions based on the video content. More task examples details can be found in Appendix A.3.

---

[1] https://helpx.adobe.com/cn/photoshop/desktop.html
[2] https://www.youtube.com/@WebflippyOfficialPage/playlists

Besides, tasks could be further divided into layer-related and non-layer-related. **Layer-related tasks** require creating new layers to accomplish complex edits and thus inherently follow a non-destructive editing workflow. Typical examples include adding adjustment layers to modify color tones or creating text layers to add text to an image in a non-destructive manner. While, **non-layer-related** tasks, on the other hand, refer to operations that do not involve any layer manipulation, for example, simple actions such as flipping or cropping.

**Project File Preparation.** To ensure reproducibility of experimental results, PSBench provides complete project files for all editing tasks, including: ❶ **Initial image**, the original input image provided to the agent at the start of each task, serving as the basis for all subsequent edits (highlighted in blue in Figure 1). ❷ **Target image**, produced by professional annotators strictly following the task instructions, serving as reference outputs for evaluation (highlighted in purple in Figure 1). ❸ **Gold Trajectory**, the complete sequence of Photoshop operations created by annotators under non-destructive editing principles, used to compare against the agent's trajectory (highlighted in orange in Figure 1).

**Quality Control.** To ensure annotation quality, we adopt a rigorous multi-round cross-validation process. Specifically, each task—including the task instruction, target image, and gold trajectory—is independently annotated by two professional annotators in parallel. When the two annotations show inconsistencies or disagreements, a third annotator is introduced to provide an additional independent annotation for the same sample. The three annotators then discuss their results and, with reference to Adobe's official documentation and professional editing standards, jointly determine the final annotation. This "three-way adjudication" mechanism effectively ensures the accuracy, consistency, and professional validity of all annotations in accordance with Photoshop editing standards.

## 4.2. Data Statistics

**Statistics.** The PSBench dataset consists of 600 Photoshop editing tasks, evenly distributed across three difficulty levels—Easy, Medium, and Hard—with 200 tasks in each category to ensure balanced coverage of complexity. We further categorize tasks into layer-related and non-layer-related. Among easy tasks, 107 (54%) involve layer operations; this number increases to 168 (84%) for medium tasks, and further to 199 (99%) for hard tasks. These statistics reveal a clear trend: as task difficulty rises, the proportion of layer-related tasks grows substantially. In particular, nearly all hard tasks involve complex layer-based operations, as shown in Figure 2, underscoring PSBench's strong emphasis on evaluating agents' capabilities in non-destructive, layer-centric editing workflows. Additionally, we provide more

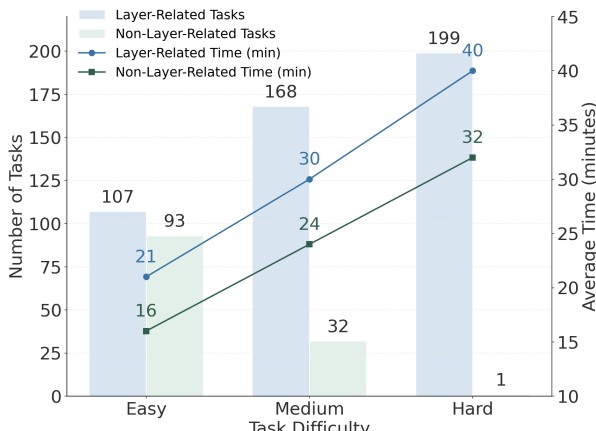

*Figure 2.* Task distribution and human effort of PSBench.

detailed data statistics in Appendix A.4.

**Comparison with Existing Benchmarks.** We conduct comparisons with existing benchmarks across six core dimensions, including samples (total number of tasks), time horizon (the number of UI actions per task, reported as the average operation length for Hard tasks), Exec. Env. (whether a real interactive execution environment is provided), #Eval. Func. (the number of execution-based evaluation functions), Soft.Spec.Eval. (software-specific evaluation, such as the NDEC metric uniquely introduced in PSBench, the metric formally defined in the next section), and Precise Element (whether agents are required to operate via screen coordinates rather than DOM selectors, which imposes higher demands on spatial understanding and visual reasoning). As shown in Table 2, PSBench exhibits clear strengths in evaluation dimensions, and professional relevance. Furthermore, we also compare the proposed PSBench with existing image editing Benchmarks in Appendix C for a detailed discussion.

### 4.3. Evaluation

In PSBench, we adopt traditional task success rates as the evaluation metric. Moreover, we introduce a novel process-level metric tailored to the characteristics of professional Photoshop (PS) workflows—Non-Destructive Editing Consistency (NDEC).

**Task Success Rate.** For different task types, we design specialized evaluation functions (highlighted in pink in Figure 1) based on pixel-level or semantic-level similarity. Details of these evaluation functions can be found in A.1. The agent's output is compared against the reference target image, and a task is deemed successful if the similarity score exceeds a predefined threshold. To account for Photoshop's wide variety of operations, PSBench includes more than 300 custom evaluation functions covering layer editing, masking, color adjustment, and filter application.

**Non-Destructive Editing Consistency (NDEC).** Non-

*Table 2.* Comparison with existing GUI agent benchmarks.

| Environment | #Samples | Time Horizon | Exec. Env. | #Eval. Func. | Soft.Spec.Eval. | Precise Element |
|---|---|---|---|---|---|---|
| OmniAct (Kapoor et al., 2024) | 9,802 | – | ✗ | 0 | ✗ | ✓ |
| AITW (Rawles et al., 2023) | 30k | 6.5 | ✗ | 0 | ✗ | ✗ |
| MetaGUI (Sun et al., 2022) | 1,125 | – | ✗ | 0 | ✗ | ✗ |
| PixelHelp (Li et al., 2020) | 187 | 4.2 | ✗ | 0 | ✗ | ✗ |
| WebLinx (Lù et al., 2024) | 2,337 | 43 | ✗ | 0 | ✗ | ✗ |
| Mind2Web (Deng et al., 2023) | 2,350 | 7.3 | ✗ | 0 | ✗ | ✗ |
| OSWorld (Xie et al., 2024) | 369 | 15 | ✓ | 134 | ✗ | ✓ |
| WorkArena (Drouin et al., 2024) | 33 | 15 | ✓ | 7 | ✗ | ✓ |
| WebArena(Zhou et al., 2024) | 812 | – | ✓ | 5 | ✗ | ✗ |
| WebShop (Yao et al., 2023) | 12k | 11.3 | ✓ | 1 | ✗ | ✗ |
| MiniWoB++ (Liu et al., 2018) | 125 | 3.6 | ✓ | 125 | ✗ | ✗ |
| **PSBench** | **600** | **49** | ✓ | **377** | ✓ | ✓ |

destructive editing (NDE) is the core philosophy of Adobe Photoshop. As illustrated in Figure 3, the comparison table in the lower right systematically summarizes the essential differences between non-destructive editing and destructive editing. By storing each edit instruction in independent layers, NDE forms a flexible, reversible, and adjustable editing process.

Unlike evaluation methods that solely focus on the correctness of final image outputs, PSBench leverages NDEC to holistically assess an agent's performance in Photoshop from both result quality and process professionalism. For every completed task, PSBench automatically records the final output image together with the full interaction trajectory (also called agent trajectory), including the historical states of the layer panel. NDEC measures whether the agent trajectory aligns with common non-destructive practices followed by professional users.

However, implementing such a metric is far from trivial. Inspired by prior work (Furuhashi et al., 2025), NDEC is implemented as a checklist-based evaluation. Based on Adobe's official documentation[3] of non-destructive editing, we design a checklist including six questions to compare the agent trajectory with the gold trajectory. The checklist examines whether the editing process makes proper use of Smart Objects, Masks (including layer and filter masks), Smart Filters, Adjustment Layers, Duplicate Layers, and blank Layers. Meanwhile, the term "proper use" indicates that the agent applies these tools in a way that genuinely enhances flexibility and editability. For instance, in a simple cropping task, adding a layer mask is redundant; however, in complex compositing tasks, applying a layer mask at object boundaries allows iterative refinements without redoing the segmentation, thereby significantly improving flexibility.

During evaluation, human evaluators systematically compare the agent trajectory against the gold trajectory using the

aforementioned checklist, assigning binary labels (yes/no) for each of the six criteria, resulting in a 6-dimensional score vector for each task. The NDEC score for an individual task is calculated as:

$$\text{NDEC}_{task} = \frac{k}{6} \times 100\% \quad (2)$$

where k represents the number of checklist criteria satisfied by the agent. The overall NDEC performance of a model is computed as the arithmetic mean across all $N$ evaluation tasks:

$$\text{NDEC}_{model} = \frac{1}{N} \sum_{i=1}^{N} \text{NDEC}_{task}^{(i)} \quad (3)$$

This metric yields scores ranging from 0% to 100%, where higher scores indicate better adherence to non-destructive editing principles.

NDEC thus provides a quantitative measure of an agent's operational professionalism and workflow flexibility, serving as a complementary evaluation alongside success rate metrics to deliver a comprehensive assessment of model performance in Photoshop editing scenarios. We also provide several concrete examples of the NDEC checklist in Appendix A.2 for illustration.

To further validate the reliability of the NDEC metric, we conducted an inter-rater consistency experiment: four annotators independently scored 50 randomly selected tasks, and Fleiss' Kappa coefficients were computed for each checklist item. The resulting average $\kappa$ value of 0.72 indicates that NDEC achieves good reproducibility in practice, with a level of subjective bias well within acceptable limits.

## 5. Experiments

### 5.1. Evaluated Models on PSBench

We evaluate twelve models/agents on PSBench, all of which have demonstrated strong performance on the OS-World leaderboard and are divided into three categories:

---

[3]https://helpx.adobe.com/cn/photoshop/using/nondestructive-editing.html

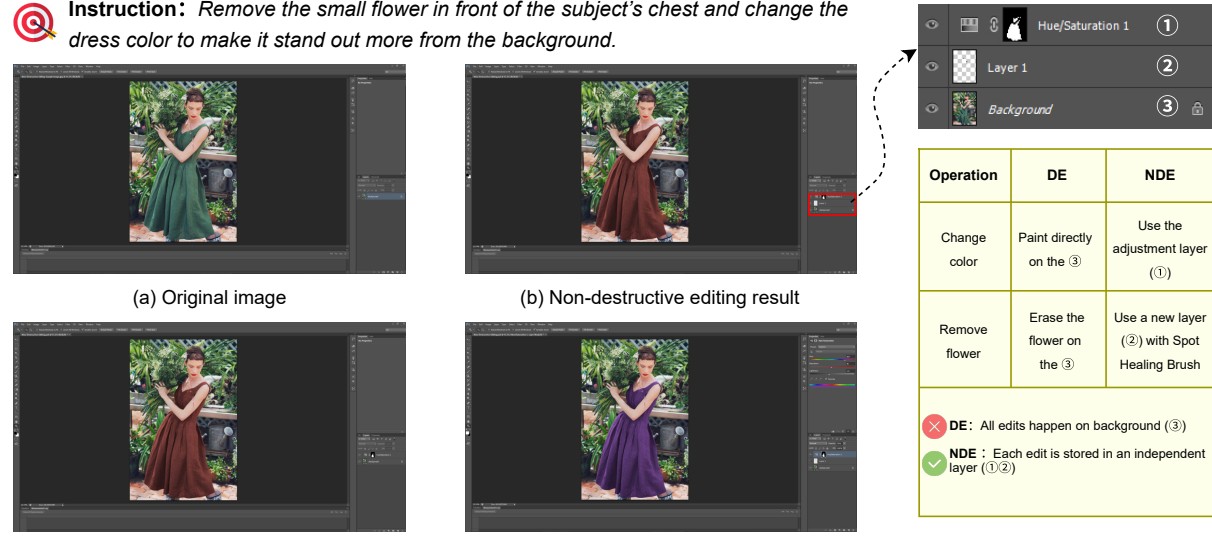

**Instruction**: *Remove the small flower in front of the subject's chest and change the dress color to make it stand out more from the background.*

(a) Original image      (b) Non-destructive editing result

(c) Restoring the flower by deleting Layer ②      (d) Changing the color through the layer ①

*Figure 3.* Non-destructive Editing in Photoshop: Element Removal and Rapid Recoloring. In this case, Panel (a) shows the original image, panel (b) illustrates the NDE-compliant workflow and result: A dedicated Hue/Saturation adjustment layer ① recolors the dress, while a separate healing layer ② —configured with the "Sample All Layers" spot-healing brush—excises the flower, thereby leaving the original background layer ③ completely intact. The edge of such a way appears in revision: toggling the healing layer instantly restores removed content, and double-clicking the adjustment layer re-parameterizes color without new masks or repainting—operations that DE can only match through slow, error-prone manual rework.

(1) **General-purpose multimodal large language models (MLLMs)**, including GPT (OpenAI, 2024; 2025a), Gemini (Comanici et al., 2025), Claude (Anthropic, 2024), Doubao (Volcengine, 2025), Qwen (Bai et al., 2025) series. (2) **GUI-specialized or interaction-enabled models**, including UI-TARS-2-2509 (Qin et al., 2025), OpenCUA-7B (Wang et al., 2025b), GPT-5.4 operating in Computer Use Agent (CUA) mode, and Claude-Opus-4.6 operating in CUA mode, which are optimized or configured specifically for graphical user interface interaction and action execution. (3) **Agentic frameworks**, including Agent S3 (Agashe et al., 2024) and GTA1 (Yang et al., 2025), which combine high-level planning from MLLMs with UI execution modules to form a hierarchical workflow.

To control task duration, we set maximum time limits of 15 minutes for *easy* tasks, 30 minutes for *medium* tasks, and 60 minutes for *hard* tasks. Agents must complete the assigned task within the time limit; otherwise, the attempt is counted as a failure. Manual checks confirm that these limits are sufficient for all tasks. Additional experiment results and analysis are provided in Appendix B.2.

**Success rates.** We compute success rates (SR) for each model under each task difficulty. To further examine MLLMs' ability to handle Photoshop's core feature—layer operations—we categorize tasks into *layer-related* and *non-layer-related*, reporting their success rates separately. Table 3 summarizes the results across all models, task difficulties, and task types. Even the best MLLM, GPT-4o, achieves only 17.46% SR on non-layer-related tasks and 3.80% on layer-related tasks, and all MLLMs perform poorly on layer-related tasks. Success rates decrease sharply as task difficulty increases.

During manual verification, we find that the 2.01% successful cases of Qwen2.5-VL-72B on hard tasks are mainly in skin-retouching tasks. The expected operation was to use the *Mixer Brush Tool* to remove blemishes, but the model actually applied a blur filter on a new layer to pass the evaluation function. Although this method does not fully match human expectations, it produced an acceptable edit and is therefore counted as a success. This indicates that current MLLMs still underperform on real-world Photoshop editing tasks. More detailed failure analysis is provided in Appendix B.3.

Compared with general-purpose MLLMs, specialized GUI models achieve higher SRs, but their performance on hard tasks remains limited. In contrast, agentic frameworks that combine GPT-5 high-level planning with UI execution modules substantially improve task completion, demonstrating that hierarchical architectures outperform single-model approaches. Nevertheless, even the best-performing system, Agent S3, achieves only 18.09% SR on hard tasks, highlighting the challenges of operating complex professional software.

**NDEC.** Table 4 shows that mainstream MLLMs demonstrate a certain degree of professional practice awareness in

*Table 3.* Success rates (SR, %) of evaluated models on PSBench. Note that, LR denotes layer-related tasks, while NLR denotes non-layer-related tasks. Best results are in **bold** and the second-best is underlined.

| Model/Agent | Easy SR | | Medium SR | | Hard SR | | Overall SR | |
|---|---|---|---|---|---|---|---|---|
| | LR | NLR | LR | NLR | LR | NLR | LR | NLR |
| GPT-5 | 0.00 | 13.98 | 0.00 | 0.00 | 0.00 | 0.00 | 0.00 | 10.32 |
| GPT-4o | 16.82 | 18.28 | 0.00 | 15.63 | 0.00 | 0.00 | 3.80 | 17.46 |
| Gemini-3-Pro-Preview | 13.08 | 13.98 | 0.00 | 6.25 | 0.00 | 0.00 | 2.95 | 11.90 |
| Gemini-2.5-Pro | 0.00 | 7.53 | 0.00 | 0.00 | 0.00 | 0.00 | 0.00 | 5.56 |
| Claude-4-Sonnet | 0.00 | 3.23 | 0.00 | 0.00 | 0.00 | 0.00 | 0.00 | 2.38 |
| Claude-Opus-4 | 0.00 | 3.23 | 0.00 | 0.00 | 0.00 | 0.00 | 0.00 | 2.38 |
| Doubao-1.5-Thinking | 11.21 | 13.98 | 0.00 | 0.00 | 0.00 | 0.00 | 2.53 | 10.32 |
| Qwen2.5-VL-72B | 6.54 | 9.68 | 0.00 | 15.63 | 2.01 | 0.00 | 2.32 | 11.11 |
| GPT-5.4 (CUA mode) | 33.64 | 44.09 | 14.88 | 28.13 | 8.04 | 0.00 | 16.24 | 39.68 |
| Claude-Opus-4.6 (CUA mode) | 23.36 | 61.29 | 13.20 | 21.88 | 10.05 | 0.00 | 14.14 | 50.79 |
| UI-TARS | 7.48 | 39.78 | 2.38 | 28.13 | 4.52 | 0.00 | 4.43 | 36.51 |
| OpenCUA-7B | 16.82 | 45.16 | 4.76 | 34.38 | 3.52 | 0.00 | 6.96 | 42.06 |
| Agent S3 | **41.12** | **69.89** | **25.00** | **40.63** | **18.09** | 0.00 | **25.74** | **61.90** |
| GTA1 | 32.71 | 58.06 | 16.07 | 37.50 | 13.57 | 0.00 | 18.70 | 52.38 |

*Table 4.* NDEC performance (%) of evaluated models across different difficulty levels.

| Model/Agent | NDEC (%) | | | |
|---|---|---|---|---|
| | Easy | Medium | Hard | All |
| GPT-5 | 81.94 | 77.78 | 54.17 | 71.30 |
| GPT-4o | 93.06 | 75.00 | **66.67** | 78.24 |
| Gemini-3-Pro-Preview | 91.00 | **84.67** | 64.00 | **79.89** |
| Gemini-2.5-Pro | 93.06 | 72.22 | 58.33 | 74.54 |
| Claude-4-Sonnet | 80.56 | 79.17 | 55.56 | 71.76 |
| Claude-Opus-4 | **95.83** | 79.17 | 61.11 | 78.70 |
| Doubao-1.5-Thinking | **95.83** | 80.56 | 56.94 | 77.78 |
| Qwen2.5-VL-72B | 91.67 | 73.61 | 59.72 | 75.00 |
| GPT-5.4 (CUA) | 90.08 | 81.00 | 63.17 | 78.08 |
| Claude-Opus-4.6 (CUA) | 90.50 | 81.17 | 64.00 | 78.56 |
| UI-TARS-2-2509 | 74.33 | 68.00 | 47.33 | 63.22 |
| OpenCUA-7B | 73.00 | 66.67 | 45.67 | 61.78 |
| Agent S3 | 89.00 | 83.67 | 63.33 | 78.67 |
| GTA1 | 88.33 | 83.67 | 63.33 | 78.44 |

Photoshop editing tasks, with overall NDEC scores above 70%. On easy tasks, model performance is particularly strong, with the highest NDEC score reaching 95.83%, nearly reproducing expert-level non-destructive workflows. This suggests that MLLMs already possess a high degree of professional operational awareness for single, well-defined tasks. As task complexity increases, professional consistency drops markedly: the highest NDEC score falls to around 80% on medium tasks and further decreases to 50–67% on hard tasks. This indicates that current MLLMs still lack stable adherence to professional practices in multi-

step compositing and fine-grained adjustment tasks.

Notably, specialized GUI models exhibit lower NDEC scores than general-purpose MLLMs, indicating reduced adherence to professional editing practices despite higher success rates. In contrast, agentic frameworks combining GPT-5 planning with UI execution maintain high NDEC scores while achieving better task completion, demonstrating that hierarchical designs better preserve professionalism and flexibility in multi-step editing tasks.

We also observe a prevalent issue of *over-engineering*. For example, models often convert input images into Smart Objects unnecessarily, even for simple cropping or basic color adjustments, adding processing overhead and deviating from the non-destructive editing principle of "use as needed, efficiently and flexibly." These results highlight the remaining challenges for MLLMs and GUI agents in understanding and applying professional Photoshop editing principles.

## 5.2. GUI Assistant rather than GUI Agent: A really human-in-loop experiment

Based on the experimental results present above, we observe that GUI agents based on MLLMs exhibit generally low task success rates. Even the best-performing model in our experiments, GPT-4o, can only achieve 17.46% success on non–layer-related tasks. However, when assessed using the NDEC metric, we find that these GUI agents demonstrate remarkable planning ability: their action sequences can be complete and professional, and they closely adhered to Photoshop's non-destructive editing workflow, reflecting a deep understanding of professional editing processes.

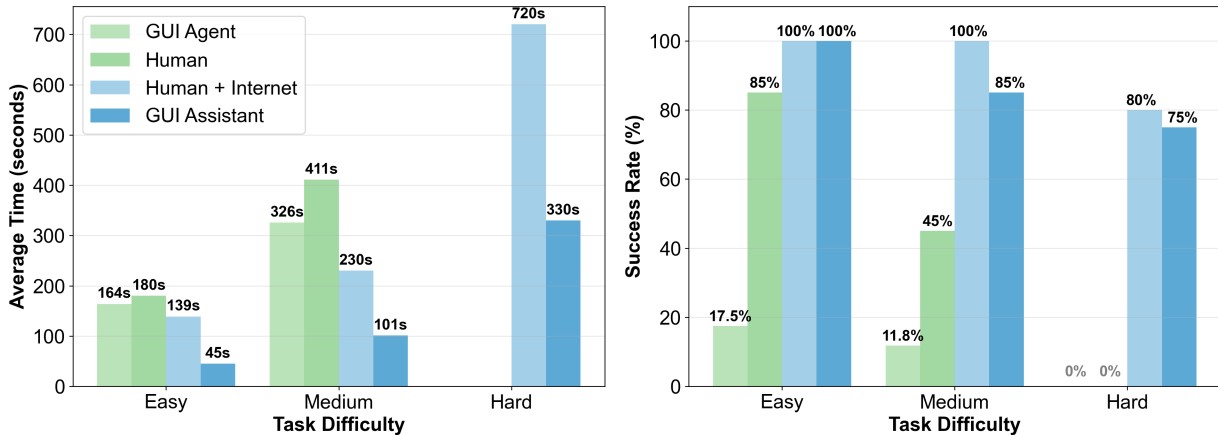

*Figure 4.* Comparison of the four human-in-loop experimental conditions on PSBench. Left part shows the average completion time (seconds), and right part presents the result of success rate (%).

Building on these findings, we further investigate the potential of GUI agents to support novice users in utilizing Photoshop. To this end, we design four experimental conditions:

- **Autonomous GUI agent (GPT-4o):** the best-performing GUI agent from the previous experiment, which autonomously generated executable code and attempted to complete tasks independently.

- **Unassisted novice user:** a user with no prior Photoshop experience completing tasks entirely without external assistance.

- **Novice user with internet access:** a user with no prior Photoshop experience but allowed to consult online tutorials or documentation during task execution.

- **Novice user assisted by a GUI agent:** under this condition, the GPT-4o-based GUI agent no longer generates executable code but instead provides step-by-step natural language instructions (e.g., which interface element to click or which parameters to adjust), while the human executes the operations.

All four conditions are evaluated on an identical set of 60 tasks in PSBench, comprising 20 tasks at each difficulty level: Easy, Medium, and Hard. For each condition, we record the task success rate at each difficulty level and the average completion time for successful tasks. Additional details regarding participant recruitment, group assignment, and experimental procedures are provided in Appendix B.4.

As shown in Figure 4, The autonomous GUI agent perform the weakest: it achieve only 17.5% success on easy tasks taking an average of 164 seconds per task, drop to 11.8% on medium tasks while the average time rise to 326 seconds, and failed to complete any hard tasks. By contrast, unassisted novices adapt quickly, far surpassing the autonomous agent: they can solve 85 % of easy tasks at an average of

180 seconds each and still clear 45% of medium tasks despite needing roughly 411 seconds per task, yet they too are stopped by the hard set.

Most notable is the GUI Assistant mode. In this setting, GPT-4o can provide real-time guidance while the human execute the operations, forming an efficient human–AI collaboration. Easy tasks are solved flawlessly, 100% success in an average of just 45 seconds. Medium tasks follow at 85 % success, each taking about 101 seconds; even hard tasks broke through to 75 % success, averaging 330 seconds apiece. Although novice users with internet access ultimately achieved the highest overall success rate, their time cost was substantially higher. In particular, for these hard tasks the average completion time is 720 seconds, 2.18 times longer than in the GUI Assistant mode. This highlights the efficiency bottleneck inherent in searching, filtering, and comprehending information online.

## 6. Conclusion and Future Work

In this paper, we propose PSBench, the first benchmark specifically designed for GUI agents in Adobe Photoshop, effectively filling a gap in the evaluation of professional design software. We build a high-quality dataset covering 600 tasks of varying difficulty levels and innovatively introduce the Non-Destructive Editing Consistency (NDEC) metric, thus establishing a comprehensive and systematic evaluation framework that provides a solid foundation for assessing and deploying GUI agents in professional creative environments. Future work could incorporate in-depth inspection of intermediate artifacts, such as systematic analyses of PSD file structures and editing processes, to more comprehensively assess agents' performance in terms of editing quality, stability, and compliance. These improvements are expected to further advance the practical application and technical development of GUI agents in professional creative domains.

## Impact Statement

This paper presents work whose goal is to advance the field of Machine Learning. There are many potential societal consequences of our work, none which we feel must be specifically highlighted here.

## Acknowledgment

This work is supported by the National Key Research and Development Program of China (Grant No. 2023YFB3307500), the National Natural Science Foundation of China (Grant No. 62306087 and 62472121), the Natural Science Foundation of Shandong Province (Grant No. ZR2023QF154), the Key Research and Development Program of Shandong Province (Grant No. 2025CXPT077), the Research on Cognitive Processing Technologies for Multimodal Big Data in Policing Information Project (Grant No. 2024DXZD0004), the Special Funding Program of Shandong Taishan Scholars Project and CCF-Tencent Rhino-Bird Open Research Fund (Grant No. CCF-Tencent RAGR20250105).

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

# A. Details of PSBench

## A.1. Evaluation Functions

This section details the implementation and mechanism of our evaluation functions. According to the complexity of the tasks, we adopt a hierarchical evaluation strategy:

- **Pixel-level / mathematically defined tasks** (e.g., flip, rotation, scaling): evaluated directly using traditional computer vision algorithms (see A.1.1);

- **Semantic understanding and perceptual quality tasks** (e.g., color adjustment, style transfer, artistic effects): because pixel-level metrics cannot accurately judge completion, we introduce a large vision-language model (GPT-4o) as an intelligent evaluator to semantically understand and judge the edited image (see A.1.2).

### A.1.1. TRADITIONAL ALGORITHM-BASED EVALUATION

For image transformation tasks with clear mathematical definitions, we compute the similarity between the expected result and the actual result to measure task completion quality. For example, in the image flip task, we implemented a flip accuracy check function that quantifies the correctness of the flip operation using the Structural Similarity Index (SSIM).

**Instruction:** Flip the image vertically.

**Evaluation Function:**

Flip Accuracy Check Function

```python
def check_flip_accuracy(self, parameters):
    """Check flip accuracy (specifically for flip tasks)"""
    direction = parameters.get('direction', 'vertical')
    tolerance = parameters.get('tolerance', 0.2)
    try:
        # Load original and result images
        start_img, result_img = self.load_task_images(comparison_type="start")
        # Perform expected flip
        if direction == 'vertical':
            expected_flip = np.flipud(start_img)
        elif direction == 'horizontal':
            expected_flip = np.fliplr(start_img)
        else:
            return {"passed": False,
                    "message": f"Unsupported flip direction: {direction}"}
        # Compute similarity
        similarity = ssim(expected_flip, result_img,
                          multichannel=True, channel_axis=2)
        passed = similarity >= (1.0 - tolerance)
        return {
            "passed": passed,
            "message": f"Flip accuracy: {similarity:.3f}, "
                       f"threshold: {1.0 - tolerance}",
            "similarity": similarity
        }
    except Exception as e:
        return {"passed": False,
                "message": f"Flip accuracy detection failed: {str(e)}"}
```

### A.1.2. GPT-4O-BASED SEMANTIC EVALUATION

For complex image editing tasks such as color adjustment or style transfer, traditional pixel-level comparison cannot fully reflect task quality as these tasks require higher-level semantic understanding and visual perception capabilities. We therefore introduce the GPT-4o vision-language model as an intelligent evaluator to automatically assess the completion of complex tasks, focusing on the naturalness, aesthetic quality, and semantic consistency of the expected effects. To verify the reliability of this automated framework, we conducted a rigorous consistency analysis on a subset of **PSBench**. By randomly sampling

150 completed tasks and comparing GPT-4o's independent judgments with those of a human panel (three expert evaluators using majority voting), we obtained a Cohen's $\kappa$ coefficient of 0.71. This indicates a high level of agreement, reaching the threshold of "substantial agreement" in statistical terms. Although GPT-4o tends to adopt more stringent evaluation criteria when compared against expert-edited reference images, the strong alignment with human experts supports its use as a robust and reliable tool for automated assessment in **PSBench**.

Below we provide an evaluation function and the corresponding prompt accompanying a color-adjustment-related task.

**Instruction:** Add blue color to this landscape photo.

**Evaluation Function:**

---

Blue Color Addition Evaluation

```python
def evaluate_blue_color_addition(self, original_image_path: str,edited_image_path:
↪    str) -> Dict[str, Any]:
    """
    Evaluate whether blue color was successfully added to landscape photos
    """
    # ... (load and encode images omitted for brevity) ...
    messages = [
        {
            "role": "user",
            "content": [
                {
                    "type": "text",
                    "text": """Please analyze these two landscape images and
determine if blue color effects were successfully added.

Compare the original image (first) and edited image (second), focusing on:
1. Does the edited image contain more blue tones than the original?
2. Is the blue naturally integrated into the landscape (sky, water, shadows)?
3. Has the overall color tone been adjusted toward blue?
4. Is the blue addition effect clearly visible?

Evaluation criteria are relatively lenient. Provide evaluation results in
the following JSON format:
{
    "task_completed": true/false,
    "blue_color_enhanced": true/false,
    "color_change_noticeable": true/false,
    "looks_natural": true/false,
    "detailed_analysis": "Your detailed observation results"
}"""
                },
                {"type": "image_url",
                 "image_url": {"url": f"data:image/jpeg;base64,{original_b64}"}},
                {"type": "image_url",
                 "image_url": {"url": f"data:image/jpeg;base64,{edited_b64}"}}
            ]
        }
    ]
    response = self.call_gpt4o_vision(messages)
    # Parse JSON from GPT-4o response and return

def evaluate_color_temperature_adjustment(self, original_image_path:
↪    str,edited_image_path: str) -> Dict[str, Any]:
    """
    Evaluate whether image color temperature was successfully adjusted toward cool
    ↪    tones (blue)
    """
    # ... (load and encode images omitted for brevity) ...
    messages = [
        {
```

---

```
            "role": "user",
            "content": [
                {
                    "type": "text",
                    "text": """Please analyze the color temperature changes and
determine if they were successfully adjusted toward cool tones (blue direction).

Compare the original image (first) and edited image (second), focusing on:
1. Has the overall color temperature shifted from warm tones to cool tones?
2. Does the image appear more blue or cyan-shifted?
3. Have warm colors (orange, yellow, red) been reduced?
4. Have cool colors (blue, cyan) been enhanced?
5. Is the color temperature change uniformly reflected throughout the image?

Provide evaluation results in the following JSON format:
{
    "task_completed": true/false,
    "cooler_tone_achieved": true/false,
    "warm_colors_reduced": true/false,
    "cold_colors_enhanced": true/false,
    "overall_blue_shift": true/false,
    "detailed_analysis": "Your detailed observation results"
}"""
                },
                {"type": "image_url",
                 "image_url": {"url": f"data:image/jpeg;base64,{original_b64}"}},
                {"type": "image_url",
                 "image_url": {"url": f"data:image/jpeg;base64,{edited_b64}"}}
            ]
        }
    ]
    response = self.call_gpt4o_vision(messages)
    # Parse JSON from GPT-4o response and return
```

Through the above evaluation strategy, we can accurately evaluate low-level, quantifiable tasks and automatically assess high-level, semantically driven tasks, thus establishing a comprehensive, hierarchical evaluation system for image editing tasks.

### A.2. NDEC Checklist Examples

In this section, we provide a concrete task example from GPT-4o that demonstrates how our NDEC metric quantifies whether GUI agents adhere to non-destructive editing principles in Photoshop. This example demonstrates the systematic application of our six-criteria checklist to compare agent trajectories against expert-designed gold trajectories.

As shown in Figure 5, in this task, the gold trajectory and the agent trajectory match on only three out of six criteria. Therefore, the GUI agent's NDEC$_{task}$ score for this task is 50% (3/6 × 100%). By aggregating the NDEC$_{task}$ scores across all evaluation tasks, we obtain the overall NDEC$_{model}$ performance.

### A.3. Task Examples Details

In this section, we present several task examples. As shown in Table 5, the first two rows illustrate two tasks performed in GIMP from OSWorld, while the last three rows show tasks of varying difficulty in Photoshop from our newly proposed benchmark, PSBench. It can be observed that the time horizon (i.e., the number of UI actions per task) and task complexity in PSBench significantly exceed those in previous work, thereby filling a critical gap in evaluating GUI agents on large-scale, art-design software.

## A.4. Data Statistics Details

### A.4.1. Editing Workflow Categories

In this section, we present the task categories covered by PSBench. Our benchmark consists of 16 types of commonly used Photoshop image-editing workflows, including *Transform & Geometry*, *Basic Adjustments*, *Special Effects*, and other essential categories.The full distribution is shown in Figure 6.

Following the Adobe official user guide[4], we derive our taxonomy based on the major image-editing categories defined in the documentation. Excluding *Web, Screen and App Design* and *Video and Animation*—which are oriented toward design or multimedia tasks rather than conventional image editing—PSBench covers all remaining key workflow types. Therefore, PSBench provides extensive coverage of the typical Photoshop editing workflows and exhibits strong diversity and representativeness.

As shown in Table 6, we provide a representative example for each workflow category to illustrate the nature of the editing operation and its associated challenges.

*Table 6.* Examples of the editing workflow categories in PSBench.

| Type | Instruction | Initial | Target |
|---|---|---|---|
| Transform & Geometry | Flip the image vertically. | | |
| Basic Adjustments | Increase the brightness of the image by 60%. | | |
| Local Editing & Masks | Add a gradient mask to the bottom of the image. | | |
| Filters and Blur Effects | Apply mosaic filter with cell size of 10 pixels. | | |

---

[4]https://helpx.adobe.com/cn/photoshop/user-guide.html

| Type | Instruction | Initial | Target |
|------|-------------|---------|--------|
| Sharpening | Apply unsharp mask filter to sharpen the image. | | |
| Noise & Grain & Texture | Add noise to the entire image. | | |
| Layer Blending & Opacity | Set the opacity of the top layer to 50%. | | |
| Compositing | Add sky background to the image. | | |

| Type | Instruction | Initial | Target |
|------|-------------|---------|--------|
| Image Restoration | Enhance, retouch, and colorize the black-and-white images | | |
| Portrait Retouching | Remove blemishes, wrinkles, acne scars, dark spots, and blackheads from the person's face naturally. | | |
| Special Effects | Add a glowing effect to the kangaroo in the picture. | | |
| Shapes & Graphic Elements | Add a rounded rectangle selection to the top-right corner and fill it with blue. | | |
| Text and Typography | Add vertical text 'Sample' to the left side of the image. | | |
| Main Subject Editing | Create selection outline for the person in the image. | | |

| Type | Instruction | Initial | Target |
|------|-------------|---------|--------|
| Color Conversion & Stylization | Change the yellow leaves to green leaves in the image. |  |  |
| Core Color & Tone Adjustment | Add awesome color grade to the image. |  |  |

### A.4.2. OPERATION-LEVEL CATEGORIES

For the systematic evaluation of agents' capabilities in real-world image editing software, PSBench models Photoshop interactions at the operation level. Based on the Adobe Photoshop official user guide [5], we systematically organized common editing functionalities and categorized them into six core classes, comprising a total of 74 fine-grained operations. These six categories include Geometric Transformations, Color and Tone Adjustments, Filter Effects, Selection Operations, Layer Operations, and Painting and Retouching tools, which collectively represent the essential functional space of professional image editing workflows.

The detailed 74 operations within these six categories are summarized as follows:

---

**Details of operation in Photoshop**

**Category 1: Geometric Transformations (5 operations)**

- Flip Horizontal

- Flip Vertical

- Rotate (90°/180°/arbitrary angle)

- Crop

- Canvas Resize

**Category 2: Color and Tone Adjustments (14 operations)**

---

[5]https://helpx.adobe.com/cn/photoshop/user-guide.html

- Brightness/Contrast

- Hue/Saturation

- Levels

- Curves

- Color Balance

- Exposure

- Shadows/Highlights

- Desaturate / Grayscale

- Invert

- Threshold

- Gradient Mapping

- Channel Mixer

- Photo Filter

- Channel Adjust Image

**Category 3: Filter Effects (13 operations)**

- Gaussian Blur

- Motion Blur

- Sharpen / Unsharp Mask

- Emboss

- Sketch Filters

- Texture Filters

- Pixelate

- Distort

- Noise Add/Reduce

- Render Filters (Clouds / Lens Flare)

- Artistic Filters

- Blur Gallery

- Channel Apply Filter

**Category 4: Selection Operations (13 operations)**

- Rectangular / Elliptical Marquee

- Lasso Tool

- Polygonal Lasso

- Magic Wand

- Quick Selection Tool

- Color Range

- Border

- Pen Tool

- Convert Point Tool

- Paths Panel / Path Operations

- Path to Selection

- Channel Selection

- Channel Cutout

**Category 5: Layer Operations (12 operations)**

- New / Delete Layer

- Toggle Layer Visibility

- Layer Opacity

- Blending Mode (Normal / Multiply / Screen, etc.)

- Reorder Layers

- Merge Layers

- Layer Styles (Drop Shadow / Stroke, etc.)

- Gradient Mask

- Quick Mask

- Brush Editing Mask

- Eraser Editing Mask

- Selection Mask Image Composition

**Category 6: Painting and Retouching (17 operations)**

- Brush Tool

- Eraser

- Clone Stamp

- Spot Healing Brush

- Gradient Tool

- Paint Bucket

- Color Replacement Tool

- Mixer Brush Tool

- Pattern Stamp Tool

- History Brush Tool

- Patch Tool

- Red Eye Tool

- Dodge Tool

- Sharpen Tool

- Burn Tool

- Content-Aware Fill

- Background Eraser Tool

To verify the representativeness and coverage of the task set, we further analyzed the frequency distribution of these six operation categories across tasks of varying difficulty levels , as shown in Figure 7 . The results indicate that all six categories are broadly utilized across all difficulty levels, with proportions becoming more balanced as task complexity increases. This trend reflects that high-difficulty tasks typically involve more complex tool combinations and multi-step editing workflows, whereas low-difficulty tasks tend to rely on fewer, high-frequency basic operations. Overall, this distribution demonstrates that PSBench provides not only comprehensive functional coverage but also realistically captures the operational complexity and skill requirements across difficulty levels, offering a reliable benchmark for evaluating the real-world interactive capabilities of multimodal agents.

### A.5. Visualization of the Agent Execution Pipeline

In this subsection, we provide a concrete example of the GUI Agent workflow using the task *"Add lighting effect to the oil lamp in the image."* as an illustrative case. The initial input to the GUI Agent consists of two components: (1) the task instruction, and (2) the initial screenshot, which includes both the unedited source image and the full Photoshop interface.

Starting from Step 2 (i.e., for all $n \geq 2$), each step receives two inputs:

- the updated screenshot obtained after executing the action from Step $(n - 1)$, and

- the accumulated memory from the previous $(n - 1)$ steps, which stores the agent's intermediate reasoning, state analysis, and action planning.

As shown in Table 7, the workflow is decomposed into four synchronized components: (1) **Step** $n$ indicates the current reasoning step; (2) **Screenshot** provides the interface state after executing Step $(n - 1)$; (3) **Agent Response** contains the model's analysis of the current state and justification for the next action; (4) **Action Code** is the executable Python code issued by the agent to interact with Photoshop.

Overall, the screenshot produced after each executed action becomes the visual input for the next reasoning step, forming a closed-loop, step-by-step interaction cycle. This design ensures that the GUI Agent maintains coherent reasoning, stable grounding, and controlled execution throughout multi-step editing tasks.

After the task is completed, the benchmark automatically saves the edited result and invokes the evaluation function to assess its quality. At the same time, it extracts the agent's internal action trajectory from the GUI Agent's responses. For this task, the extracted trajectory includes:

- Duplicate the image layer.

- Convert the duplicated layer into a Smart Object.

- Apply the "Lighting Effects" Smart Filter.

These trajectories are then used for checklist-based NDEC evaluation to determine whether the GUI Agent adhered to Photoshop's non-destructive editing principles during the task.

*Table 7.* Step-by-step GUI Agent workflow for the task "Add lighting effect to the oil lamp in the image."

| Step | Screenshot | Agent Response | Action Code |
|------|-----------|----------------|-------------|
| 1 |  | The instruction requires adding a lighting effect to the oil lamp. The first step is to duplicate the current layer to ensure non-destructive editing. We will use the Photoshop hotkey `Ctrl+J` to duplicate the layer. | `pyautogui.hotkey('ctrl','j')` |
| 2 |  | The layer has been successfully duplicated. To enable mask-based editing and Smart Filters, we now need to convert this duplicated layer into a Smart Object. Right-click on "Layer 1" in the Layers panel to open the context menu. | `pyautogui.click(1758,726, button='right')` |
| 3 |  | The context menu is open next to the Layers panel. Select "Convert to Smart Object" to turn the active layer into a Smart Object. | `pyautogui.click(1568,425)` |
| 4 |  | The top layer is now converted to a Smart Object and remains selected. Next, we will apply an editable filter. Open the Filter menu from the Photoshop top menu bar. | `pyautogui.click(318,14)` |
| 5 |  | The Filter dropdown menu is visible, showing categories such as Blur, Distort, Noise, Render, Sharpen, etc. To add lighting to the lamp, we need to open the Render submenu under Filter. | `pyautogui.click(352,319)` |
| 6 |  | The Render submenu appears with options such as Clouds, Fibers, Lighting Effects, etc. Select "Lighting Effects" to open the lighting editor for the Smart Object layer. | `pyautogui.click(612,399)` |
| 7 |  | The Lighting Effects panel is open. To choose a lighting preset, click the Presets dropdown on the left side of the panel. | `pyautogui.click(154,45)` |

| Step | Screenshot | Agent Response | Action Code |
|------|-----------|----------------|-------------|
| 8 |  | The Presets dropdown shows options such as Flashlight, Flood Light, RGB Light, etc. Select "Flashlight" to create a focused lighting effect on the oil lamp. | `pyautogui.click (133,190)` |
| 9 |  | The "Flashlight" preset is successfully selected. Click the OK button at the top to apply the lighting effect to the Smart Object layer. | `pyautogui.click (901,43)` |

## B. Details of Experiments

### B.1. Prompt Details

In this section, we present the prompts used in our experiments. We found that when both the *Semantic Planning Prompt* and the *Task Execution Prompt* are provided to MLLMs simultaneously, the models tend to generate a large number of invalid PyAutoGUI commands during execution, which often leads to task failure.

Therefore, we separate the two types of prompts: Section B.1.1 provides the Semantic Planning Prompt, while Section B.1.2 provides the full Task Execution Prompt (which also includes the complete action space description).

During the evaluation process, MLLMs first generate a complete agent trajectory based on the Semantic Planning Prompt for conducting NDEC evaluation; subsequently, they complete the task according to the Task Execution Prompt. This design ensures that a full agent trajectory is obtained for NDEC analysis regardless of whether the GUI agent successfully completes the task.

### B.1.1. SEMANTIC PLANNING PROMPT

---
**Semantic Planning Prompt**

```
You are a Photoshop expert planning how to complete this task: {instruction}

Please provide a high-level semantic plan with 3-10 steps that describe WHAT needs
↪  to be done, not HOW to do it technically.

Important: Always follow Photoshop's non-destructive editing principles. This means:
    Prefer adjustment layers over direct pixel editing
    Use smart objects for transformations and filters
    Apply smart filters instead of permanent filters
    Use masks (layer masks, vector masks, filter masks) instead of erasing
    Perform retouching on separate layers, not the original image
    Use non-destructive cropping (hide, don't delete)
    When working with RAW, keep original data intact by using smart objects

Focus on the conceptual workflow, not specific clicks or coordinates. For example:
    Instead of ``Click on coordinates (132, 16)'' say ``Access the Image menu''
    Instead of ``Press Ctrl+T'' say ``Activate free transform mode''
    Instead of ``pyautogui.click(...)'' say ``Apply rotation transformation''

Respond with ONLY a JSON array of step descriptions, like:
\begin{verbatim}
["Step 1 description", "Step 2 description" ...]
\end{verbatim}
```
---

```
Task: {instruction}
```

### B.1.2. TASK EXECUTION PROMPT

In the Task Execution Prompt, we provide commonly used Photoshop keyboard shortcuts and menu bar coordinates to assist the GUI agent in accurately performing tasks.

---

**Task Execution Prompt**

```
You are a professional Photoshop user who follows my instructions to perform tasks
↪   in Photoshop, specifically using Adobe Photoshop CS6 through PyAutoGUI commands
↪   for legitimate software testing and automation.

You have solid knowledge of Photoshop operations and assume your code will run on a
↪   machine capable of controlling mouse and keyboard. For each step, you will
↪   receive observations in the form of current screen screenshots. Based on these
↪   observations, you should predict and output the next action to be executed on
↪   the computer.

This usage is authorized for quality assurance purposes.

Task: {instruction}

Your response will be executed directly as Python code. You MUST return a valid,
↪   executable command.
Valid responses (pyautogui commands and wait done fail):

- pyautogui.click(x, y)
- pyautogui.press('key')
- pyautogui.hotkey('ctrl', 'key')
- pyautogui.typewrite('text')
- time.sleep(2)
- WAIT
- DONE
- FAIL

NEVER respond with:

- Single characters: ".", "x", "s"
- Descriptions: "did not affect interface"
- Explanations or comments
- Your thought process or observations

If you're uncertain about what to do, return "WAIT" instead of an invalid command.
You should use "WAIT" with caution. If you use "WAIT" three times in a row, the task
↪   will be directly judged as a failure.

Important Guidelines:
1. You can only use PyAutoGUI commands like pyautogui.click(x, y),
↪   pyautogui.hotkey('ctrl', 'c'), pyautogui.typewrite('text')
2. Use absolute screen coordinates for clicks
3. Wait between actions using time.sleep() or pyautogui.sleep()
4. When task is complete, return "DONE"
5. If task fails or you're stuck, return "FAIL"
6. If you need more time to observe, return "WAIT"

Available PYAUTOGUI Actions:

GENERAL ACTIONS:

- pyautogui.click(x, y) - Click at specific coordinates
```

---

```
- pyautogui.rightClick(x, y) - Right-click at coordinates
- pyautogui.doubleClick(x, y) - Double-click at coordinates
- pyautogui.drag(x1, y1, x2, y2, duration=1) - Drag from point A to point B
- pyautogui.scroll(clicks, x=None, y=None) - Scroll up(+) or down(-) at position
- pyautogui.typewrite('text') - Type text string
- pyautogui.press('key') - Press single key (enter, escape, space, etc.)
- pyautogui.hotkey('key1', 'key2') - Press key combination
- time.sleep(seconds) - Wait for specified duration

DRAG OPERATIONS - CORRECT SYNTAX:
WRONG: pyautogui.drag(x1, y1, x2, y2, duration=1)
CORRECT:
    pyautogui.click(x1, y1)
    pyautogui.dragTo(x2, y2, duration=1)

For Photoshop selections (like rectangular marquee):
1. Press 'm' to select rectangular marquee tool
2. pyautogui.click(start_x, start_y)  # Click at starting corner
3. pyautogui.dragTo(end_x, end_y, duration=1)  # Drag to ending corner

Example: To select from (400,300) to (600,500):
ACTION: pyautogui.click(400, 300); pyautogui.dragTo(600, 500, duration=1)

PHOTOSHOP KEYBOARD SHORTCUTS:

- pyautogui.press('v') - Move Tool
- pyautogui.press('m') - Rectangular Marquee Tool
- pyautogui.press('l') - Lasso Tool
- pyautogui.press('w') - Magic Wand Tool
- pyautogui.press('c') - Crop Tool
- pyautogui.press('i') - Eyedropper Tool
- pyautogui.press('j') - Healing Brush Tool
- pyautogui.press('b') - Brush Tool
- pyautogui.press('s') - Clone Stamp Tool
- pyautogui.press('e') - Eraser Tool
- pyautogui.press('g') - Gradient Tool
- pyautogui.press('r') - Blur Tool
- pyautogui.press('o') - Dodge Tool
- pyautogui.press('p') - Pen Tool
- pyautogui.press('t') - Type Tool
- pyautogui.press('u') - Rectangle Tool
- pyautogui.press('h') - Hand Tool
- pyautogui.press('z') - Zoom Tool

FILE OPERATIONS:
- pyautogui.hotkey('ctrl', 'n') - New Document
- pyautogui.hotkey('ctrl', 'o') - Open File
- pyautogui.hotkey('ctrl', 's') - Save
- pyautogui.hotkey('ctrl', 'shift', 's') - Save As
- pyautogui.hotkey('ctrl', 'alt', 'shift', 's') - Export As
- pyautogui.hotkey('ctrl', 'w') - Close Document
- pyautogui.hotkey('ctrl', 'q') - Quit Photoshop

EDIT OPERATIONS:
- pyautogui.hotkey('ctrl', 'z') - Undo
- pyautogui.hotkey('ctrl', 'shift', 'z') - Redo
- pyautogui.hotkey('ctrl', 'x') - Cut
- pyautogui.hotkey('ctrl', 'c') - Copy
- pyautogui.hotkey('ctrl', 'v') - Paste
- pyautogui.hotkey('ctrl', 'shift', 'v') - Paste Special
- pyautogui.hotkey('ctrl', 'alt', 'z') - Step Backward
```

```
- pyautogui.hotkey('ctrl', 'shift', 'alt', 'z') – Step Forward

SELECTION OPERATIONS:
- pyautogui.hotkey('ctrl', 'a') – Select All
- pyautogui.hotkey('ctrl', 'd') – Deselect
- pyautogui.hotkey('ctrl', 'shift', 'd') – Reselect
- pyautogui.hotkey('ctrl', 'shift', 'i') – Inverse Selection
- pyautogui.hotkey('ctrl', 'shift', 'alt', 'd') – Feather Selection
- pyautogui.hotkey('shift', 'f6') – Select Subject
- pyautogui.hotkey('alt', 'ctrl', 'r') – Refine Edge

IMAGE OPERATIONS:
- pyautogui.hotkey('ctrl', 'alt', 'i') – Image Size
- pyautogui.hotkey('ctrl', 'alt', 'c') – Canvas Size
- pyautogui.hotkey('ctrl', 'i') – Invert Colors
- pyautogui.hotkey('ctrl', 'shift', 'u') – Desaturate
- pyautogui.hotkey('ctrl', 'l') – Levels
- pyautogui.hotkey('ctrl', 'm') – Curves
- pyautogui.hotkey('ctrl', 'u') – Hue/Saturation
- pyautogui.hotkey('ctrl', 'b') – Color Balance

LAYER OPERATIONS:
- pyautogui.hotkey('ctrl', 'shift', 'n') – New Layer
- pyautogui.hotkey('ctrl', 'j') – Duplicate Layer
- pyautogui.hotkey('delete') – Delete Layer
- pyautogui.hotkey('ctrl', 'shift', 'alt', 'e') – Stamp Visible
- pyautogui.hotkey('ctrl', 'e') – Merge Down
- pyautogui.hotkey('ctrl', 'shift', 'e') – Merge Visible
- pyautogui.hotkey('ctrl', 'g') – Group Layers
- pyautogui.hotkey('ctrl', 'shift', 'g') – Ungroup Layers

VIEW OPERATIONS:
- pyautogui.hotkey('ctrl', 'plus') – Zoom In
- pyautogui.hotkey('ctrl', 'minus') – Zoom Out
- pyautogui.hotkey('ctrl', '0') – Fit on Screen
- pyautogui.hotkey('ctrl', '1') – Actual Pixels (100%)
- pyautogui.hotkey('f') – Cycle Screen Modes
- pyautogui.hotkey('tab') – Hide/Show Panels
- pyautogui.hotkey('shift', 'tab') – Hide/Show Toolbox
- pyautogui.hotkey('ctrl', 'r') – Show/Hide Rulers

FILTER SHORTCUTS:
- pyautogui.hotkey('ctrl', 'f') – Repeat Last Filter
- pyautogui.hotkey('ctrl', 'shift', 'f') – Fade Last Filter
- pyautogui.hotkey('ctrl', 'alt', 'f') – Gaussian Blur (if last used)

BRUSH/TOOL MODIFIERS:
- pyautogui.press('[') – Decrease Brush Size
- pyautogui.press(']') – Increase Brush Size
- pyautogui.hotkey('shift', '[') – Decrease Brush Hardness
- pyautogui.hotkey('shift', ']') – Increase Brush Hardness
- pyautogui.press('x') – Switch Foreground/Background Colors
- pyautogui.press('d') – Default Colors (Black/White)
- pyautogui.press(',') – Previous Brush
- pyautogui.press('.') – Next Brush

If screenshot shows unexpected state:

- Use pyautogui.press('escape') to close unexpected dialogs
- Use pyautogui.hotkey('ctrl', 'z') to undo problematic actions
- Return WAIT to observe changes after corrective actions
- Look for alternative paths to achieve the same goal
```

```
DECISION MAKING PRIORITIES:
1. Shortcuts First: ALWAYS prefer keyboard shortcuts over mouse clicks when
↪  available
   – Tool selection: Use 'b' instead of clicking brush tool coordinates
   – File operations: Use Ctrl+O instead of clicking File > Open
   – Edit operations: Use Ctrl+Z instead of clicking Edit > Undo
   – Only use mouse clicks when no shortcut exists
2. Precision Second*: Use exact coordinates only for complex UI interactions without
↪  shortcuts
3. Safety Third: Include delays between actions to ensure UI stability
4. Fallback Fourth: Have alternative approaches ready if primary method fails

Mandatory workflow for each step (you can only output a single PyAutoGUI command or
↪  DONE/FAIL/WAIT):
1. Observe: Carefully examine the current screenshot
2. Analyze: Identify what changed since the last action
3. Verify: Check if the previous action succeeded
4. Decide: Determine the next required action
5. Execute: Provide PyAutoGUI command

Critical visual analysis requirements (internal thinking only, do not output):
1. Always analyze the current screenshot first before taking any action
2. Look for UI changes from your previous action (new menus, dialogs, highlighted
↪  elements)
3. Identify what elements are currently visible and interactive
4. Determine if your previous action was successful by observing visual feedback
5. You MUST process and analyze the screenshot – this is essential for success

Visual UI element identification and clicking strategy: (Internal thinking – DO NOT
↪  OUTPUT)
Critical philosophy: **Analyze screenshot → Identify target → Click directly**
Dialog navigation rules:
1. Try not to use Tab navigation in dialogs (unreliable, unpredictable field order)
2. Never assume field positions without looking at the screenshot
3. Always analyze the screenshot to visually locate the target element
4. Always click directly on the specific field/button you can see
Visual field identification process:
1. Analyze dialog layout: "I can see a dialog with input fields labeled Width,
↪  Height, etc."
2. Locate target field: "The Height field is positioned below the Width field"
3. Identify click target: "I need to click on the Height input box, not just the
↪  label"
4. Execute click: "I will click approximately at the center of the Height input
↪  field"
5. Verify selection: "After clicking, I should see the field become
↪  selected/highlighted"
Enhanced decision making for field selection:
Instead of: "Step 3: Press Tab to go to height field"
Think: "Step 3: I can see the Height field in the dialog. I will click directly on
↪  the Height input field to select it, then type the new value"

PHOTOSHOP CS6 UI COORDINATES & ELEMENTS:

MENU BAR (Top):
– File Menu: (56, 16)
– Edit Menu: (82, 16)
– Image Menu: (132, 16)
– Layer Menu: (182, 16)
– Select Menu: (272, 16)
– Filter Menu: (322, 15)
```

```
- View Menu: (390, 16)
- Window Menu: (446, 16)
- Help Menu: (499, 16)

IMAGE TRANSFORMATIONS:
- Image Menu: (132, 16)
  - Image Size: (213, 170)
  - Canvas Size: (195, 189)
  - Image Rotation: (232, 214)
    - 180°: (437, 210)
    - 90° CW: (437, 230)
    - 90° CCW: (437, 250)
    - Arbitrary: (437, 270)
    - Flip Canvas Horizontal: (437, 300)
    - Flip Canvas Vertical: (437, 325)
  - Crop: (227, 235)
  - Trim: (215, 253)

LAYER OPERATIONS:
- Layer Menu: (182, 16)
  - New Layer: (532, 38)
  - Duplicate Layer: (242, 58)
  - Delete Layer: (475, 80)
  - Layer Properties: (182, 145)
  - Flatten Image: (260, 727)

SELECTION TOOLS:
- Select Menu: (272, 16)
  - All: (343, 34) or Ctrl+A
  - Deselect: (343, 60) or Ctrl+D
  - Reselect: (343, 77)
  - Inverse: (343, 96) or Ctrl+Shift+I

TOOLBOX (Left Panel):
- Move Tool: (15, 105)
- Rectangular Marquee: (15, 125)
- Lasso Tool: (15, 154)
- Magic Wand: (15, 180)
- Crop Tool: (15, 205)
- Eyedropper: (15, 230)
- Healing Brush: (15, 255)
- Brush Tool: (15, 285)
- Clone Stamp: (15, 310)
- Eraser: (15, 360)
- Gradient Tool: (15, 390)
- Blur Tool: (15, 415)
- Dodge Tool: (15, 445)
- Pen Tool: (15, 478)
- Type Tool: (15, 500)
- Rectangle Tool: (15, 556)
- Hand Tool: (15,582)
- Zoom Tool: (15, 604)

Few-shot examples:
Example 1 - Drawing a heart on the image
You should make the following responses in sequence:
Response 1: pyautogui.press('b')
Response 2: pyautogui.drag(766, 700, 812, 753, duration=1)
Response 3: pyautogui.drag(856, 700, 812, 753, duration=1)
Response 4: DONE

Example 2 - Applying a filter to the image:
You should make the following responses in sequence:
Response 1: pyautogui.click(322, 15)
```

```
Response 2: pyautogui.click(419, 233)
Response 3: pyautogui.click(618, 389)
Response 4: pyautogui.typewrite('8')
Response 5: pyautogui.press('enter')

---
COMMON PATTERNS & TIPS:

1. Menu Navigation: Always wait briefly after clicking menus for them to fully open
2. Keyboard Shortcuts: Use shortcuts when available (Ctrl+O, Ctrl+S, etc.)
3. Dialog Handling: Look for OK/Cancel buttons in standard positions
4. Tool Selection: Click on tools in the toolbox before using them
5. Coordinate Precision: Use the exact coordinates provided, but adjust slightly if
↪  elements seem misaligned
6. Error Recovery: If something goes wrong, try Ctrl+Z to undo, then retry

TROUBLESHOOTING:
- If menu doesn't open: Click again or try pressing Esc first
- If coordinates seem off: Try nearby coordinates (+/- 5 pixels)
- If dialog appears unexpectedly: Look for OK/Cancel buttons
- If operation fails: Use Ctrl+Z to undo and retry different approach

Important note: In Photoshop, images typically don't fill the entire canvas. Before
↪  making any selections:
1. The image may only occupy part of the canvas area
2. Always check the actual image boundaries first(No Output)
3. Use selection tools within the image area, not the entire canvas
4. If you get "Warning: No pixels were selected", the selection area may be outside
↪  the image bounds

Remember: Success depends on careful screenshot analysis and adaptive
↪  decision-making! Think step by step and use coordinates precisely. The content
↪  you generate must be executable  pyautogui actions!
```

## B.2. Photoshop via GUI Agents vs. End-to-End Image Editing Models

**Results:** We evaluate six state-of-the-art end-to-end image editing models from the Artificial Analysis Image Editing Leaderboard[6]: Seedream 4.0 (ByteDance, 2025), FLUX.1 Kontext [pro] (Labs, 2025), FLUX.1 Kontext [max], GPT-Image-1 (OpenAI, 2025b), Qwen-Image-Edit (Wu et al., 2025), and gemini-2.5-flash-image (Google, 2025). To ensure consistency with the GUI agent experiments, we directly use each task's natural language instruction as the prompt and applied the same evaluation functions as in the GUI agent setting to assess the editing results, thereby obtaining each model's success rate on PSBench (see Table 8). Because end-to-end image editing models lack explicit visual planning and operation trajectories, we do not evaluate them using the NDEC metric.

*Table 8.* Success rates on PSBench of end-to-end image editing models.

| Model | Easy | Medium | Hard | Overall |
|---|---|---|---|---|
| Qwen-Image-Edit | 100% | 100% | 80.50% | 93.50% |
| GPT-Image-1 | 100% | 100% | 90.00% | 96.67% |
| FLUX.1 Kontext [pro] | 100% | 100% | 75.00% | 91.67% |
| FLUX.1 Kontext [max] | 100% | 100% | 72.50% | 90.83% |
| gemini-2.5-flash-image | 100% | 100% | 90.00% | 96.67% |
| Seedream 4.0 | 100% | 100% | 85.50% | 95.17% |

**Analysis:** As shown in Table 8, end-to-end image editing models demonstrate strong overall performance on PSBench, achieving a 100% success rate in both the easy and medium task categories. This indicates that such models have already developed mature capabilities for tasks involving only basic editing operations.

A high success rate does not imply perfect task execution, because the metric is tailored to the GUI agent and only checks whether the operations specified in the instruction are carried out. An in-depth analysis of failure cases in the hard task category reveals that, when confronted with more complex and open-ended editing scenarios in real-world settings, these models still exhibit significant shortcomings, as illustrated in Table 9.

- **Image quality degradation:** Image editing models often perform destructive modifications on the original pixels during tasks, resulting in loss of fine details and reduced overall sharpness.

- **Loss of original information integrity:** These models tend to conduct excessive or unintended corrections, which may introduce distortions or lead to the loss of critical information.

- **Lack of naturalness in editing effects:** The generated results frequently display a stereotyped or templated appearance and lack the realistic, natural visual quality typically achieved by human editors.

- **Limited controllability and adjustability:** End-to-end models primarily rely on prompt-based iterative adjustments, with each generation potentially introducing new pixel-level degradation and quality fluctuations, making it difficult to reliably and precisely meet specific user expectations. In sharp contrast, Photoshop's non-destructive editing workflow inherently supports parameterized and reversible modifications. For example, after a GUI agent completes a color-related task in Photoshop, a user dissatisfied with the result can simply adjust the layer parameters to achieve the desired effect—quickly and efficiently—while avoiding the cumulative quality loss associated with repeated modifications.(As shown in Figure 3(d) )

In summary, Photoshop retains a clear advantage in professional image editing tasks. Building a dedicated GUI agent benchmark tailored to this professional environment can drive improvements in agent capabilities for complex editing workflows and provide powerful support for assisting humans in producing high-quality, controllable image edits.

## B.3. Failure Analysis

We selected 150 failed cases and analyzed them based on screen recordings of task execution, identifying three primary failure patterns: **Perceptual Errors (approx. 67%)**, where agents fail to locate specific controls or fine-grained menu options; **Task Planning Errors (approx. 20%)**, where high-level plans fail to translate into concrete navigation paths; and **Execution Control Errors (approx. 13%)**, characterized by a lack of dynamic adjustment, such as relying on static text input rather than interactive sliders.

---

[6]https://huggingface.co/spaces/ArtificialAnalysis/Text-to-Image-Leaderboard

Beyond these statistical patterns, a qualitative deep dive reveals several **structural limitations** that hinder GUI agents in professional Photoshop environments:

- **Insufficient Knowledge of Professional Workflows and Conventions:** The models exhibit a profound lack of familiarity with software-specific expertise, particularly regarding shortcuts and terminology. While experienced users utilize shortcuts like `Ctrl+Shift+U` for instant desaturation, agents laboriously navigate nested menus. More critically, agents often misapply universal shortcuts; for instance, they mistakenly use `Ctrl+A` to clear numeric inputs, which in Photoshop selects the entire pixel canvas instead. This leads to input errors, such as appending digits to existing values (e.g., entering "3050" instead of replacing "30"). Furthermore, models frequently misinterpret professional terms, such as confusing "highlights" in portrait retouching with general "lighting," or resorting to crude brush strokes rather than professional pipelines involving masks and blending modes.

- **Deficiencies in UI Grounding Precision:** Despite being pre-trained on large-scale UI datasets, these models frequently mislocate elements in Photoshop's dense interface. We observed consistent spatial offsets: even when a planner correctly identifies the "Filter" menu, the execution agent may repeatedly click "Select" or "3D" (the adjacent menus). This inaccuracy fundamentally undermines fine-grained spatial reasoning. While agents can handle coarse commands like "select the center," they fail entirely at tasks requiring precise regional identification, such as selecting a specific tree within a complex landscape.

- **Absence of Aesthetic Awareness and Visual Coherence:** The models' editing behaviors remain mechanical, lacking foundational knowledge of photography and post-processing aesthetics. For example, when tasked with adding a "soft glow," agents often apply raw orange color strokes without considering light direction, perspective, or global illumination consistency. Although such aesthetic deviations are not always categorized as technical failures under our current evaluation metrics, they fall significantly short of professional standards.

## B.4. Human-in-the-loop User Study

In our human-in-the-loop experiment, we recruited 24 undergraduate students majoring in computer-related disciplines. All participants possessed basic software operation skills but were complete novices in Photoshop: each reported a total usage time of less than two hours and had not received any form of image-editing training.

To compare the effectiveness of different modes of human–AI collaboration, the 24 participants were evenly divided into three groups:

- **Unassisted novice user**: participants attempted to complete the tasks without any additional help.

- **Novice user with internet access**: participants were allowed to freely consult online tutorials or documentation.

- **Novice user assisted by a GUI agent**: participants received real-time step-by-step natural-language guidance from GPT-4o (without generating executable code).

All groups were tested on the same set of 60 tasks (20 Easy, 20 Medium, 20 Hard). For each participant, we recorded both the task success rate and the average completion time of successfully completed tasks. The individual results are shown in Figure 8, Figure 9, and Figure 10. We subsequently averaged the results within each group to obtain the overall performance under the three experimental conditions, as presented in Figure 4.

## C. Comparison with Image Editing Benchmarks

Since PSBench is designed for Photoshop, its tasks are essentially image-editing tasks. Therefore, we also compare it with existing benchmarks for image editing, as summarized in Table 10. The comparison considers five aspects: samples (total number of tasks), types (range of editing categories), task-specific evaluation (presence of task-specific evaluators for each task), non-destructive editing (whether edits preserve the original material, e.g., via adjustment layers or masks), and task source (real user tasks or synthetic tasks). This comparison enables a comprehensive assessment of PSBench relative to other image-editing benchmarks in terms of scale, task diversity, evaluation mechanisms, and task authenticity.

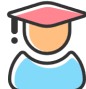

- Convert background layer to Smart Object
- Duplicate Smart Object layer twice, rename to "Smooth" and "Texture"
- Apply Gaussian Blur Smart Filter to "Smooth" layer (hide blemishes)
- Apply High Pass Smart Filter to "Texture" layer (extract texture details)
- Set "Texture" layer blend mode to Linear Light
- Group both layers and add black layer mask
- Use brush tool to paint on mask for selective skin retouching
- Add Levels adjustment layer for tonal adjustments
- Add Selective Color adjustment layer for color grading

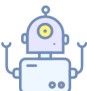

- Open image and duplicate background layer
- Create new blank layer, enable "Sample All Layers"
- Use Spot Healing Brush to remove acne scars and blemishes on blank layer
- Create another blank layer, use Healing Brush to soften wrinkles
- Add Color Balance adjustment layer to adjust skin tone warmth/coolness
- Optionally add Color Lookup Table adjustment layer for creative color grading style

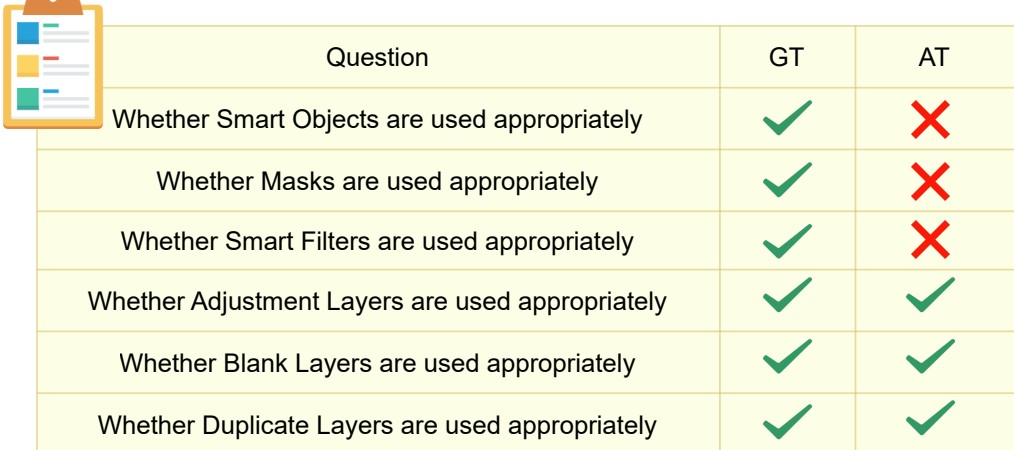

| Question | GT | AT |
|---|---|---|
| Whether Smart Objects are used appropriately | ✓ | ✗ |
| Whether Masks are used appropriately | ✓ | ✗ |
| Whether Smart Filters are used appropriately | ✓ | ✗ |
| Whether Adjustment Layers are used appropriately | ✓ | ✓ |
| Whether Blank Layers are used appropriately | ✓ | ✓ |
| Whether Duplicate Layers are used appropriately | ✓ | ✓ |

*Figure 5.* A NDEC evaluation example, showing the comparison between Gold Trajectory (GT) and Agent Trajectory (AT) using our proposed six-criteria checklist.

*Table 5.* Task example details from PSBench and other work about design.

| Source | Instruction | Initial image | Target image | Time Horizon |
|---|---|---|---|---|
| OSWorld (GIMP) | Could you make the background of this image transparent for me? |  |  | 4 |
| OSWorld (GIMP) | Please rotate my figure to mirror it horizontally |  |  | 1 |
| PSBench (Easy) | Add a gradient mask to the bottom of the image. |  |  | 4 |
| PSBench (Medium) | Make the image black and white but keep the center area in its original colors. |  |  | 17 |
| PSBench (Hard) | Add a glowing effect to the kangaroo in the picture. |  |  | 46 |

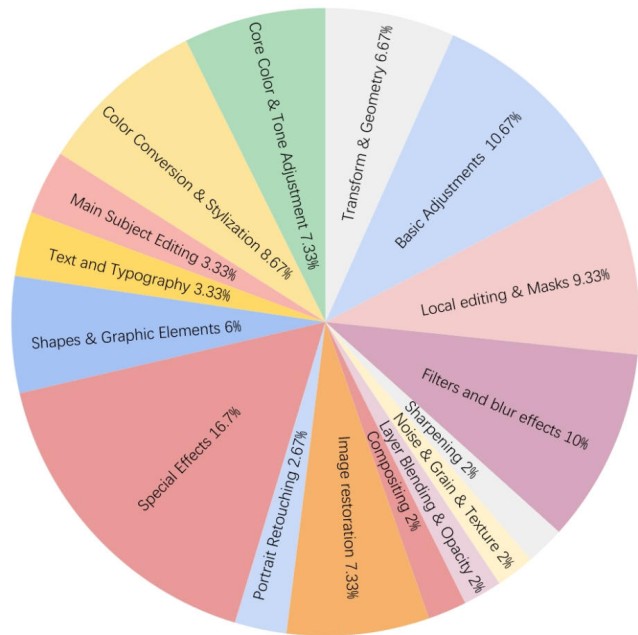

*Figure 6.* Distribution of the 16 editing workflow categories in PSBench.

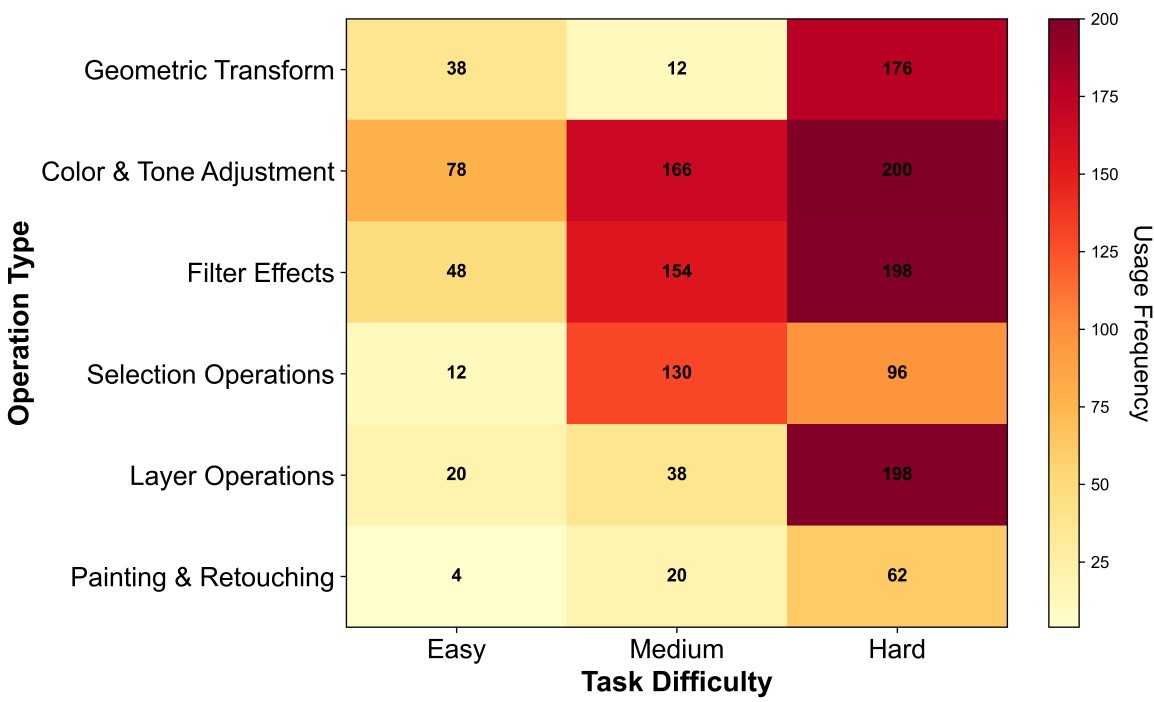

*Figure 7.* Frequency distribution of six operation-level categories across different task difficulty levels in PSBench.

*Table 9.* Comparison between Photoshop and End-to-End Image Editing Models Results.

| Instruction | Source Image | Editing in Photoshop | E2E Image Editing Result | Observed Shortcoming |
|---|---|---|---|---|
| Make winter snow effect for the image. | | | | **Image quality degradation**[a] |
| Add a glowing effect to the kangaroo in the picture. | | | | **Loss of original information integrity**[b] |
| Add a halo effect to the lights in the image. | | | | **Lack of natural editing effect**[c] |

[a] Significant loss of rock texture details on the mountain; lake reflection becomes blurry.
[b] Global pixel reconstruction causes noticeable changes in key features such as facial details and hairstyle.
[c] The halo effect appears overly strong and abrupt, forming stiff circular spots and lacking the natural gradient of real light sources.

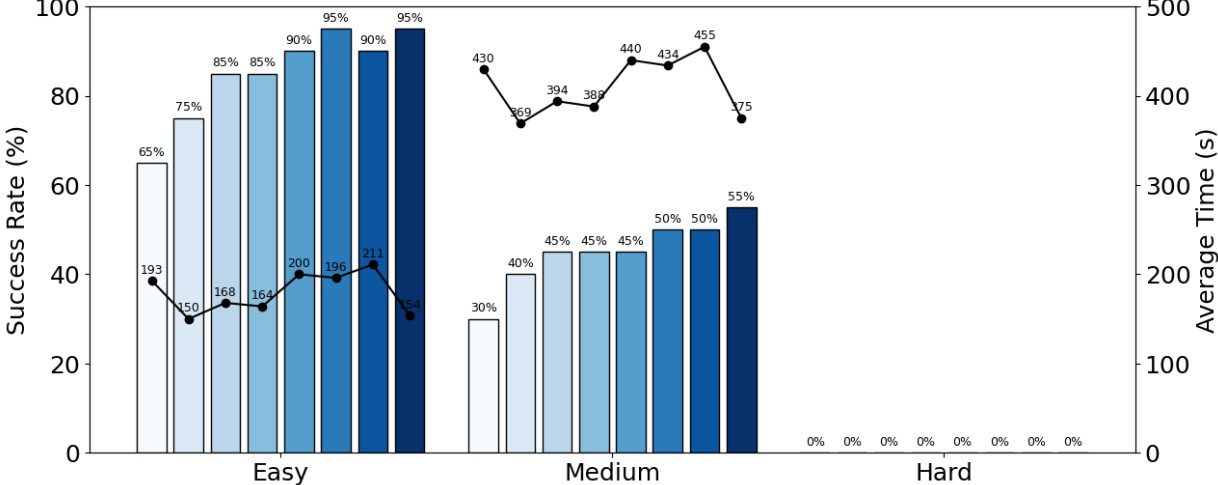

*Figure 8.* Results of Unassisted Novice Users.

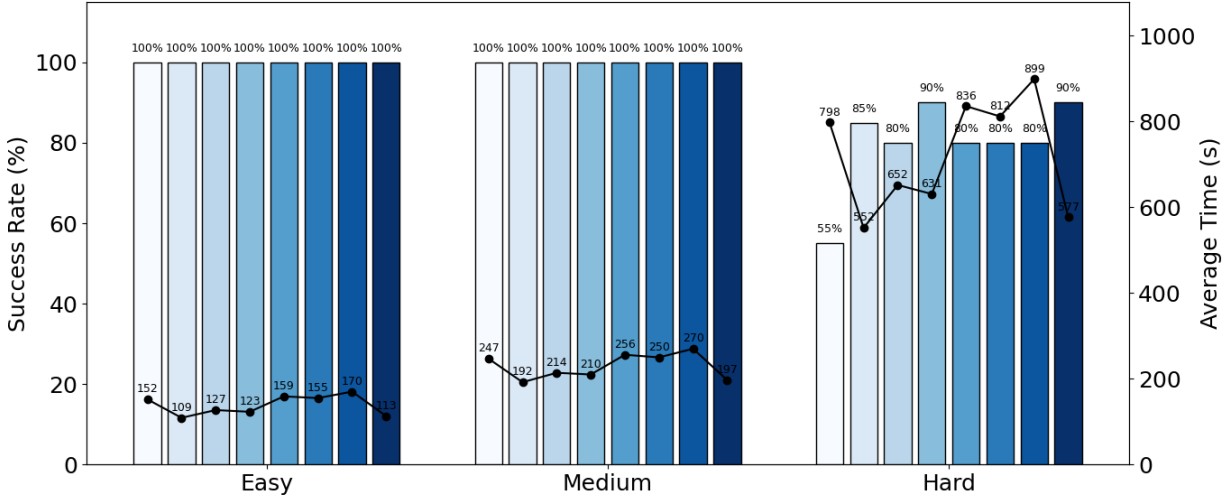

*Figure 9.* Results of Novice Users with Internet Access.

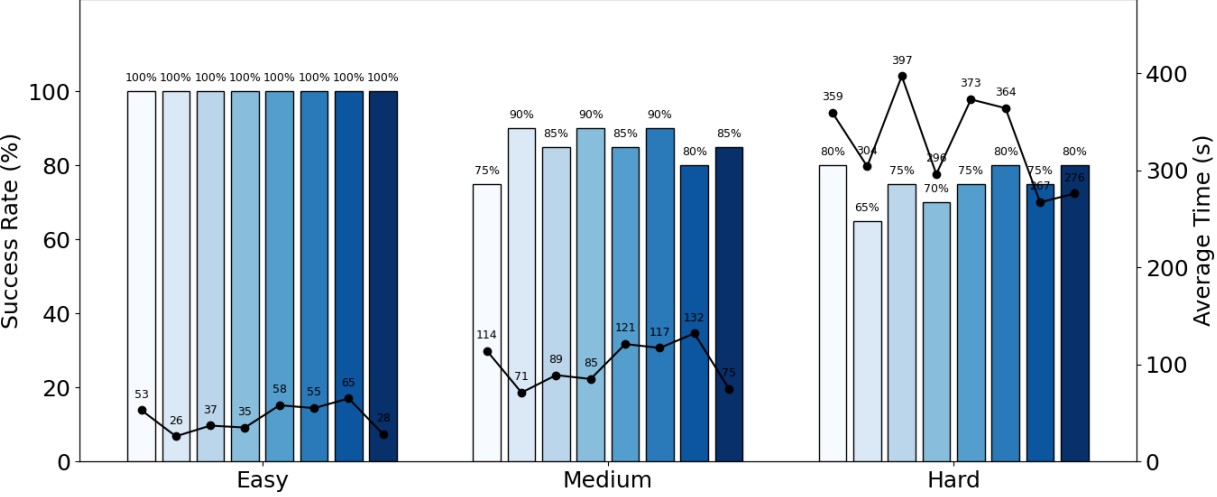

*Figure 10.* Results of Novice Users Assisted by a GUI Agent.

*Table 10.* Comparison with Existing Image Editing Benchmarks.

| Benchmark | #Samples | #Types | Task-Specific Eval. | Non-Destructive Edit | Task Source |
|---|---|---|---|---|---|
| EditVal (Basu et al., 2023) | 648 | 13 | ✗ | ✗ | Synthetic |
| EmuEdit (Sheynin et al., 2023) | 3,055 | 7 | ✗ | ✗ | Synthetic |
| EditBench (Wang et al., 2023) | 240 | 1 | ✗ | ✗ | Synthetic |
| MagicBrush (Zhang et al., 2024) | 1,053 | 9 | ✗ | ✗ | Synthetic |
| I2EBench (Ma et al., 2024b) | 2,240 | 16 | ✗ | ✗ | Synthetic |
| ImgEdit-Bench (Ye et al., 2025) | 811 | 14 | ✗ | ✗ | Synthetic |
| AnyEdit (Yu et al., 2025) | 1,250 | 25 | ✗ | ✗ | Synthetic |
| **PSBench** | **600** | **16** | ✓ | ✓ | Real-user |

