# OpenReview forum: "PSBench: Editing Image via GUI Agents in Photoshop"
_ICML.cc/2026/Conference — ICML 2026 regular_

### Official Review · Reviewer_mHAu · 2026-02-24

**Soundness:** 3
**Presentation:** 3
**Significance:** 2
**Originality:** 2
**Overall Recommendation:** 4
**Confidence:** 3

**Summary:**

PSBench is an agentic workflow image editing benchmark that is designed to evaluate agents while operating using Adobe Photoshop. The operation level modeled in this benchmark is the GUI operations like mouse movement and keyboard actions. The 600 human-annotated Photoshop tasks spanning three difficulty levels (easy, medium, hard), with tasks drawn from official tutorials.
The benchmark evaluates 12 state-of-the-art MLLMs, GUI-specialized models, and hierarchical agentic frameworks. Results show very low autonomous success rates on complex tasks (e.g., Agent S3 achieves \~18% on hard tasks), but relatively high NDEC scores for MLLMs, indicating strong planning awareness despite weak execution.

**Compliance With Llm Reviewing Policy:**

Affirmed.

**Final Justification:**

As I wrote in my comments to authors -  I still keep my score due to the fact that there is no research novelty, but the study is interesting and dataset can be useful.

**Key Questions For Authors:**

Generalization Beyond Photoshop CS6 - How sensitive is PSBench to the specific Photoshop version?
Would UI layout changes significantly alter performance?
If the benchmark is brittle to version updates, its long-term utility may be limited.

**Limitations:**

Discussed. Apart from the fact that no research novelty (it is more a dataset paper maybe?).

**Strengths And Weaknesses:**

Strengths:
1.	While prior works have suggested browser-aligned agentic workflows, operating on a software like Photoshop is not the same and raises the bar in agentic capabilities to longer (harder) tasks which also require higher precision for GUI. Like autonomous vehicle operating today on the same protocols, we human use – the roads, traffic signs, steering wheel, etc. So does this benchmark. It makes sense to align agents with the same interpretable GUI of a software and to operate on the mouse and keyboard like a gas paddle and a steering wheel.

2.	The authors propose the Non-Destructive Editing Consistency (NDEC) metric, designed to assess whether agents adhere to Photoshop’s non-destructive editing philosophy. This is a universal best-practice that goes beyond Photoshop.

3.	The benchmark design is technically well specified.

4.	The presentation is clear and mostly polished, with minor editorial improvements needed.


Weaknesses:
1.	This benchmark is application-specific to Photoshope. Its scope is limited to Adobe software, and more concerning is how well will it mature or become obsolete. Meaning, how fast will its software versioning diverge from the frozen dataset. When Adobe move buttons will it break PSBench? It is also unclear what is the feature-coverage of these 600 examples for the targeted Photoshop version.

2.	There is no Prior works section. SOTA works are mentioned through the Introduction section and following sections.

3.	This submission does not include any novel method. It only contributes a benchmark for evaluating Adobe Photoshop tasks and how well SOTA methods perform on it today. Having even a baseline method at the very least reassures the dataset is training-ready. That said, I feel like this benchmark has potential to be the vision-language-action milestone for other methods to build upon.

4.	The human-in-the-loop experiment sample size (60 tasks) is moderate but not large and statistical significance tests are not reported.

---

> ### Author Rebuttal · Authors · 2026-03-31
>
> >W1&Q1
>
> We thank the reviewer for raising critical questions regarding environmental reproducibility and UI stability. Regarding your concerns about the potential impact of UI element position variations and Photoshop version updates on benchmark results, we provide clarification from both our methodological principles and supplementary experiments. Due to the character limit of the rebuttal, we have provided a more detailed discussion in our response to Reviewer oiMi's Q1 & W1; we kindly invite you to refer to that section.
>
> Additionally, regarding your question "It is also unclear what is the feature-coverage of these 600 examples for the targeted Photoshop version," we have provided a detailed explanation in Appendix B.4. This section systematically analyzes the functional coverage of PSBench from two levels: editing workflow categories and operation-level taxonomy.
>
> >W2
>
> Regarding the section on prior works, due to space limitations, we have organized the related work in the appendix (see Appendix A: Related Work).
>
> >W4
>
> Regarding the reviewer's concerns about the lack of statistical significance testing, we have supplemented the corresponding statistical analysis here.
>
> First, we conducted a descriptive statistical analysis of the experimental results across different difficulty levels and auxiliary conditions. We calculated the mean and standard deviation of task completion time and success rate for each group, as summarized in the table below. Overall, both Internet assistance and the GUI Assistant outperformed the Unassisted condition in terms of task success rate and efficiency, with this difference being more pronounced in medium- and high-difficulty tasks.
>
> Building on this, we employed the two-tailed Mann–Whitney U test to analyze the significance of key inter-group differences. The results indicate that, compared to the Unassisted group, Internet assistance yielded a significant improvement in task success rate ($p<0.01$); similarly, the GUI Assistant also demonstrated a significant improvement in success rate over the Unassisted group ($p<0.01$).
>
> Further comparing the efficiency difference between the two auxiliary modalities, the results show that for Hard tasks, the completion time of the GUI Assistant group was significantly lower than that of the Internet-assisted group ($p<0.001$). In terms of the magnitude of efficiency improvement, the average completion time of the GUI Assistant on Hard tasks was approximately $0.46\times$ that of the Internet group, representing an efficiency speedup of roughly $2.2\times$.
>
> |Difficulty|Group|Success(%)|Time(s)|
> |---|---|---|---|
> |Easy|Unassisted|85±10|179±22|
> |Easy|Internet|100±0|138±22|
> |Easy|GUI|100±0|45±14|
> |Medium|Unassisted|45±8|410±29|
> |Medium|Internet|100±0|230±28|
> |Medium|GUI|85±5|101±20|
> |Hard|Internet|80±11|720±132|
> |Hard|GUI|75±6|329±46|

---

> > ### Author Rebuttal · Reviewer_mHAu · 2026-04-03
> >
> > Thanks you for the rebuttal and clarifications.
> > I still keep my score due to the fact that there is no research novelty, but the study and dataset can be useful.

---

> > > ### Author Response · Authors · 2026-04-03
> > >
> > > We sincerely thank the reviewer for the positive evaluation of our work. We greatly appreciate the valuable suggestions, which have helped us improve the clarity and overall quality of the manuscript.

---

### Official Review · Reviewer_oiMi · 2026-03-13

**Soundness:** 3
**Presentation:** 2
**Significance:** 3
**Originality:** 3
**Overall Recommendation:** 5
**Confidence:** 2

**Summary:**

This paper presents PSBench, a Photoshop-specific benchmark and evaluation framework for GUI agents, rather than a new GUI-agent method itself. It builds a real interactive Photoshop environment with 600 human-annotated tasks, a coordinate-based action space, more than 300 task-specific evaluators, and a software-specific process metric, Non-Destructive Editing Consistency (NDEC), to assess whether agents follow professional non-destructive editing workflows. The experiments show that current GUI agents still perform poorly on complex Photoshop tasks, while human-in-the-loop “GUI assistant” use is substantially more practical.

**Compliance With Llm Reviewing Policy:**

Affirmed.

**Final Justification:**

I recommend accept (5). The paper addresses an important and underexplored problem, and I find the benchmark contribution meaningful and likely useful to the community. The rebuttal resolved my main concerns sufficiently, so I increased my score, although some practical limitations still remain.

**Key Questions For Authors:**

1. UI determinism: What exact OS / screen resolution / scaling / language settings are assumed? How sensitive is success rate to small UI layout shifts?

2. NDEC cost: What is the per-task human effort for NDEC scoring, and do you envision a scalable evaluation protocol (e.g., partial annotation, active sampling, automated heuristics)?

**Limitations:**

yes

**Strengths And Weaknesses:**

Strengths:

1. Strong benchmark motivation + clear gap: Photoshop is a genuinely complex, professional, multi-state UI where current GUI agents are expected to struggle, and the paper makes that case convincingly.

2. High-quality task design at realistic difficulty: 600 tasks, stratified difficulty, and coverage of 74 fine-grained operations across 6 operation categories, plus workflow diversity.

3. Thorough baseline suite and analysis: includes multiple MLLMs, specialized GUI models, and hierarchical agent frameworks; includes failure analysis and a useful human-in-the-loop study.

Weaknesses:

1. Reproducibility / legal accessibility concerns: the environment relies on a “portable” Photoshop CS6 installation. It is unclear how the benchmark can be reproduced broadly and legally (and across OS/UI scaling settings) without tightly specified setup artifacts.

2. Evaluation robustness: some tasks rely on semantic evaluators and the paper notes that evaluation can be “passed” with unintended operations (e.g., blur filter instead of mixer brush). This suggests the evaluation functions may still be exploitable or incomplete for some workflows.

3. NDEC scalability: NDEC is checklist-based and requires human evaluation; this may limit scalability for future leaderboard-style benchmarking unless a cheaper proxy or sampling strategy is proposed.

---

> ### Author Rebuttal · Authors · 2026-03-31
>
> >W1&Q1
>
> We sincerely thank the reviewer for raising these critical questions regarding environmental reproducibility and UI stability.
>
> All experiments used Windows (1920×1080 resolution, 100% scaling, English). We will include these in the final version.Furthermore, our testing confirms that PSBench can run stably on both Windows and macOS. It is worth noting that since Adobe Photoshop does not offer an official Linux version, PSBench does not support the Linux platform at this time.
>
> Regarding the reviewer's concern about whether variations in the positions of UI elements affect the success rate, we provide clarification from both our methodological principles and supplementary experiments:
>
> (1) Methodological Principles (An example is provided in Appendix B.5).
>
> In PSBench, the GUI agent does not perform operations by "memorizing" fixed UI coordinates. Instead, at each step, the agent parses the interface based on the current screenshot. It identifies the semantic functions of UI elements and performs UI grounding to determine the locations of the corresponding controls. Therefore, the model relies on visual understanding and semantic matching rather than fixed coordinate positions. This design ensures that the benchmark possesses a degree of robustness against changes in UI layout.
>
> (2) Supplementary Experiments.
>
> To further verify this, we conducted two sets of additional experiments:
>
> - **Experiment A**: We swapped the positions of the Layers panel and the Tools panel, and re-tested the success rates of two models. (Due to time constraints, we have initially tested two representative models; the remaining baseline results will be completed in the future). The success rates(%) are shown in the table below:
>
> |GUI Agent|Easy SR (LR/NLR)|Medium SR (LR/NLR)|Hard SR (LR/NLR)|Overall SR (LR/NLR)|
> |---|---|---|---|---|
> |GPT-5|0/13.98|0/0|0/0|0/10.32|
> |UI-TARS|8.41/37.63|2.38/25.00|4.52/0|4.64/34.13|
>
> - **Experiment B**: We changed the Photoshop version from CS6 to Photoshop 2020 (released roughly 8 years apart) and re-evaluated the model performance.The success rate(%) are shown in the table below:
>
> |GUI Agent|Easy SR (LR/NLR)|Medium SR (LR/NLR)|Hard SR (LR/NLR)|Overall SR (LR/NLR)|
> |---|---|---|---|---|
> |GPT-5|0/15.05|0/0|0/0|0/11.11|
> |UI-TARS|9.35/36.56|2.98/28.13|4.02/0|4.85/34.13|
>
> As shown in the supplementary experiments, the fluctuation in the Overall Success Rate does not exceed approximately 3% under either UI layout changes or software version updates. Further manual verification indicates that the vast majority of tasks maintained consistent outcomes before and after these changes: tasks that were originally successful generally remained successful in the new setups, while failed tasks mostly remained failed. Only a very small number of tasks showed changes in results. It is worth pointing out that such slight fluctuations also occasionally occur during repeated runs under identical UI and version settings. This is primarily caused by the inherent decision-making instability of the GUI agent itself, rather than differences in UI layout or versions.
>
> Based on the above analysis, we believe that PSBench demonstrates strong robustness against reasonable UI layout changes and cross-version discrepancies.
>
> >W2
>
> We acknowledge that for certain complex editing workflows, the LLM-as-judge evaluation approach cannot achieve 100% accuracy comparable to human review. In image editing tasks, different operational paths can sometimes produce similar visual results. During our manual verification process, we reviewed screen recordings of over a thousand task executions and identified only a single such instance—specifically, the case mentioned by the reviewer in the manuscript (Line 317).It is worth noting that this case passed the evaluation not because the evaluation function was "misled," but because the final edited result was visually acceptable and successfully achieved the goal of skin blemish removal. Our evaluation primarily focuses on whether the final editing effect meets the intended goal, rather than strictly enforcing that the model must use a specific tool.To further verify the reliability of our automatic evaluation framework, we also conducted a consistency analysis (Line 758). The results demonstrate that, despite a few very rare edge cases, the semantic evaluator as a whole still provides robust and reliable evaluation signals.
>
> >W3&Q2
>
> We thank the reviewer for pointing out the concern regarding the scalability of NDEC. We agree that since NDEC adopts a checklist-based manual evaluation approach, it does incur a certain amount of labor cost. In our annotation experiments, it took a skilled annotator an average of approximately 190 seconds to inspect a complete agent trajectory and complete the six NDEC scoring items. We will further explore more scalable evaluation protocols in future versions to support larger-scale benchmark evaluations.

---

> > ### Author Rebuttal · Reviewer_oiMi · 2026-04-04
> >
> > The authors have provided clarifications on reproducibility, UI layout stability, and the scalability of their evaluation. However, concerns about the practical applicability of the benchmark and potential issues with evaluation remain. There is still a need to explore scalable evaluation protocols and enhance the benchmark's general usability for broader testing.

---

> > > ### Author Response · Authors · 2026-04-05
> > >
> > > We sincerely thank the reviewer for this valuable suggestion. To enhance the scalability of our evaluation, we further explored an LLM-as-a-judge automated evaluation scheme to verify whether large language models can serve as a close proxy for human evaluation. Specifically, we selected execution trajectories of four representative models across different difficulty levels and employed two state-of-the-art LLMs (Claude Sonnet 4.6 and GPT 5.4) as judges to automatically score the NDEC. The results are summarized below:
> > >
> > > **Table 1: NDEC (%) with Claude-Sonnet-4.6 as the judge**
> > >
> > > | Model | Easy | Medium | Hard | All |
> > > |-------|------|--------|------|-----|
> > > | Claude_4_Sonnet | 84.67 | 72.33 | 64.33 | 73.78 |
> > > | Gemini-3-Pro-Preview | 86.00 | 79.67 | 62.67 | 76.11 |
> > > | GPT-4o | 89.00 | 71.33 | 68.33 | 76.22 |
> > > | Qwen2.5-VL-72B | 84.67 | 72.67 | 62.67 | 73.33 |
> > >
> > > **Table 2: NDEC (%) with GPT-5.4 as the judge**
> > >
> > > | Model | Easy | Medium | Hard | All |
> > > |-------|------|--------|------|-----|
> > > | Claude_4_Sonnet | 79.67 | 73.33 | 60.67 | 71.22 |
> > > | Gemini-3-Pro-Preview | 85.33 | 82.67 | 71.00 | 79.67 |
> > > | GPT-4o | 84.33 | 71.67 | 63.67 | 73.22 |
> > > | Qwen2.5-VL-72B | 82.67 | 76.67 | 66.00 | 75.11 |
> > >
> > > Comparing these results with the human evaluation in Table 4 (Line 348) of our paper, we observe that both Claude and GPT, as judges, consistently capture the downward trend in model performance as task difficulty escalates from Easy to Medium to Hard. This aligns perfectly with the overall pattern observed in human evaluation. Furthermore, across different difficulty settings, the deviation in NDEC scores between the LLM judges and human evaluators is strictly kept within $10\%$, demonstrating that LLM-as-a-judge can effectively replicate the general trends of human scoring.
> > >
> > > These results indicate that LLM-as-a-judge holds strong potential as a scalable proxy for NDEC human evaluation, significantly reducing human annotation costs while maintaining evaluation reliability. Due to the limited time available for this rebuttal, this experiment serves as a preliminary verification on a subset of representative models. In our future work, we plan to expand the scale of this evaluation and systematically investigate the stability and generalization of this automated evaluation protocol across a wider range of models and task settings.

---

### Official Review · Reviewer_rzWc · 2026-03-26

**Soundness:** 3
**Presentation:** 3
**Significance:** 3
**Originality:** 3
**Overall Recommendation:** 4
**Confidence:** 4

**Summary:**

This paper introduces PSBench, a GUI agent benchmark for image editing in Adobe Photoshop. It consists of 600 human-annotated tasks across three difficulty levels. It evaluates multiple models/agents alongside a human-in-the-loop study.

**Compliance With Llm Reviewing Policy:**

Affirmed.

**Final Justification:**

The rebuttal addressed some of the concerns, but without directly responding to several of them:

W1.1: The rebuttal explained that the unsharp masks are fixed and predefined, but did not respond directly to the question: "If the agent applies a reasonable but different unsharp mask than the annotator's, would it fail the SSIM check?"

W2: The rebuttal mentioned "restarting Photoshop" as an environment reset, with "manual review of screen recordings across thousands of task runs" as a guarantee of the benchmark results. However, whether Photoshop could accumulate state across sessions or being cross-task contaminated is still not addressed by the rebuttal.

Q2: The authors added experiments on gpt-5.4 and claude-opus-4-6, but not Anthropic's Computer Use agent or OpenAI's CUA. The authors did not explain why they were omitted.

A GUI benchmark work in the context of complex domain-specific scenarios like photo editing is **indeed intriguing and brings unique insights to the community**. But given that several concerns are unaddressed, there is no room for rating this work any higher.

**Key Questions For Authors:**

1. GPT-4o, used as the evaluation judge for semantic tasks, was retired from the OpenAI API in February 2026. What is the plan for maintaining and reproducing the benchmark going forward? Has the evaluation been validated with any successor model?
2. Anthropic's Computer Use Agent and OpenAI's CUA are absent from the evaluation. Why they are omitted?
3. Why was Photoshop CS6 (released more than 10 years ago) chosen over more recent versions?

**Limitations:**

yes

**Strengths And Weaknesses:**

### Strengths

1. The benchmark fills a clear gap in GUI agent evaluation. Existing benchmarks focus on general-purpose software or web environments with relatively simple interactions. Professional creative software like Photoshop presents genuinely harder challenges, and this is a well-motivated benchmark direction.

2. The human-in-the-loop experiment is practical and compelling, reframing the value of current GUI agents from full automation toward interactive assistance.

3. The annotation effort is substantial, with multi-round cross-validation and three-way adjudication.

### Weaknesses

1. Many tasks in the benchmark are vaguely defined "open questions", and one concern could be that the evaluation criteria may not align well with these vague definitions. Some concrete examples:
   - "Apply unsharp mask filter to sharpen the image" does not specify radius, amount, or threshold. If the agent applies a reasonable but different unsharp mask than the annotator's, would it fail the SSIM check?
   - "Add awesome color grade to the image" is subjective. What counts as "awesome"? The evaluation here relies on GPT-4o's judgment, which may not agree with other reasonable interpretations. One argument could be: in this specific example, the target image might be oversaturated, and the teal-and-orange look might not be a good choice.
   - "Brighten the eyes in the portrait" checks a specific region. But what if the agent brightens the entire image? The eyes would also be brighter, so would it pass? Conversely, what if the agent produces a better result with smooth blending at region boundaries but fails the pixel-level comparison?

2. It's unclear whether Photoshop's execution environment is kept clean and consistent between tasks. Photoshop accumulates state across sessions (recent files, default settings, and possibly layout changes), and without explicit resets, cross-task contamination could affect reproducibility.

---

> ### Author Rebuttal · Authors · 2026-03-31
>
> >W1
>
> We thank the reviewer for these valuable comments. We acknowledge that in an image editing environment like Photoshop, some tasks do possess a certain degree of open-endedness. Regarding the specific examples you mentioned, we clarify them as follows:
>
> (1) Actually, the parameters for the Unsharp Mask filter are explicitly specified in our task definition. When displaying the task list in the Appendix, we simplified the description to "Apply unsharp mask filter to sharpen the image" to maintain a clean and readable table layout, thus omitting the specific parameters in the table cells. In the actual benchmark configuration, these parameters are fixed and predefined, ensuring that the task output is deterministic and can be stably evaluated via metrics like SSIM.
>
> (2) We appreciate the reviewer's suggestion. We agree that the phrasing "awesome color grade" is indeed subjective and may lead to varied interpretations. In the final version, we will rewrite this task to correspond to a more explicit color adjustment goal (e.g., specifying a concrete tone or color adjustment operation) to reduce ambiguity and improve the consistency and reproducibility of the evaluation.
>
> (3) For such local editing tasks, we do not rely on pure pixel-level comparisons. Instead, we adopt an LLM-as-a-Judge evaluation approach and explicitly require the model to judge whether the edit is applied only to the specified region in the evaluation prompt. We conducted dedicated tests for the two scenarios the reviewer raised: (i) If the agent brightens the entire image globally, the evaluation model will recognize that the editing scope does not match the task requirement and rate it as a failure; (ii) If the agent brightens only the eye area with a more natural transition at the boundary, the evaluation model will still rate it as a success, even if the result has subtle pixel-level differences from the reference image.
>
> In summary,to guarantee evaluation determinism, we have explicitly specified key parameters in our task design as much as possible and utilized the LLM-as-a-Judge method for semantic-level evaluation in appropriate scenarios. Furthermore, we conducted a human consistency test (Line 760), which demonstrates that our automatic evaluation method achieves high agreement with human expert judgments, thereby validating its reliability in PSBench.
>
> >W2
>
> We appreciate the concern regarding environment consistency and reproducibility. PSBench features an explicit reset mechanism: upon completing each task, the Photoshop process is terminated and restarted for the next. This ensures task independence. All open images and intermediate edits from the previous task are closed to prevent cross-task state carryover. Furthermore, our manual review of screen recordings across thousands of task runs revealed no failures caused by state accumulation.We further discussed the impact of minor UI element variations on PSBench's robustness. Due to character limits, this experiment and its results are detailed in our response to Reviewer oiMi (Q1 & W1, Experiment A). We kindly refer the reviewer to that section for details.
>
> >Q1
>
> We thank the reviewer for this key point. While we used GPT-4o for certain semantic tasks, PSBench does not depend on any specific LLM. Here, GPT-4o acts solely as an LLM-based evaluator governed by explicit prompts and rules rather than proprietary behaviors. Thus, our framework is model-agnostic: any capable LLM can drop in using the identical prompts and protocols. To ensure long-term maintainability and reproducibility, we will validate results with newer models and report cross-evaluator consistency in the final version.
>
> >Q2
>
> We thank the reviewer for the suggestion. We have added these two sets of experiments, and results are as follows:
>
> |GUI Agent|Easy SR (LR/NLR)|Medium SR (LR/NLR)|Hard SR (LR/NLR)|Overall SR (LR/NLR)|NDEC(Easy)|NDEC(Medium)|NDEC(Hard)|NDEC(All)|
> |---|---|---|---|---|---|---|---|---|
> |gpt-5.4|33.64/44.09|14.88/28.13|8.04/0|16.24/39.68|90.08|81.00|63.17|78.08|
> |claude-opus-4-6|23.36/61.29|13.20/21.88|10.05/0|14.14/50.79|90.50|81.17|64.00|78.56|
>
> >Q3
>
> We thank the reviewer for this question. We chose Photoshop CS6 primarily because it functionally covers the majority of core image editing workflows (such as layers, selections, filters, etc.), which fully supports the main operation types involved in PSBench.Meanwhile, this version is free from redundant cloud-based AI features that could interfere, facilitating precise task design during the benchmark construction phase and in-depth error attribution analysis during the experimental phase.
>
> To further verify the applicability of PSBench across different Photoshop versions, we conducted a supplementary cross-version experiment. Due to character limits, this experiment and its results are detailed in our response to Reviewer oiMi (Q1 & W1, Experiment B). We kindly direct the reviewer there for details.

---

> > ### Author Rebuttal · Reviewer_rzWc · 2026-04-03
> >
> > The rebuttal did not respond to several concerns (see final justification), but the core contribution of the paper is still valid. Therefore the score would be kept unchanged.

---

> > > ### Author Response · Authors · 2026-04-05
> > >
> > > We sincerely thank the reviewer for the insightful feedback. In the first round of rebuttal, due to strict character limits, we were unable to fully elaborate on the experimental naming conventions for Q2. Regarding the reviewer's concern in the Final Justification that "the authors added experiments on gpt-5.4 and claude-opus-4-6, but not Anthropic's Computer Use agent or OpenAI's CUA. The authors did not explain why they were omitted," we wish to provide the following clarification:
> > >
> > > Our previously added experiments on gpt-5.4 and claude-opus-4-6 essentially encompass the "OpenAI CUA" and "Anthropic's Computer Use agent" referenced by the reviewer. It is important to clarify that Anthropic's and OpenAI's CUA are not standalone systems separate from their backbone models. Instead, they are "Agent modes" enabled through specific protocols, such as Anthropic's computer-use protocol and OpenAI's computer tool interface. In our experiments:
> > >
> > > - **Anthropic CUA**: In the claude-opus-4-6 experiments, we explicitly enabled the `computer-use-2025-11-24` beta header [1]. This not only invokes the model but also activates its officially defined "Computer Use" capability, allowing it to generate precise pixel-level coordinate mappings and keyboard/mouse control sequences.
> > >
> > > - **OpenAI CUA**: For the gpt-5.4 experiments, we integrated the official computer response tool [2]. This interface is the core technical mechanism defining OpenAI's CUA (also known as Operator or Computer-Using Agent) mode, facilitating autonomous control over complex GUIs through a closed-loop feedback mechanism within the designated action space.
> > >
> > > The reviewer's feedback highlighted that simply using the backbone model names (e.g., gpt-5.4) may lead readers to misinterpret our evaluations as mere "multimodal understanding tests" rather than "closed-loop agentic control tests." To eliminate this ambiguity, we will update the experimental labels in the revised manuscript to be more explicit, such as gpt-5.4 (CUA mode), to more clearly reflect that the computer interaction capabilities were enabled.
> > >
> > > **References**
> > >
> > > [1] https://platform.claude.com/docs/en/agents-and-tools/tool-use/computer-use-tool
> > >
> > > [2] https://developers.openai.com/api/docs/guides/tools-computer-use

---

### Decision · Program_Chairs · 2026-04-30

**Decision:**

Accept (regular)

**Comment:**

Final rating: 5: Accept / 4: Weak Accept / 4: Weak Accept

The paper introduces PSBench, the first GUI agent benchmark specifically designed for professional image editing in Adobe Photoshop. The dataset contains 600 human-annotated tasks categorized into three difficulty levels, covering 74 fine-grained operations across six categories. Reviewers appreciated the benchmark's focus on a genuinely complex, professional software environment, filling a clear gap left by existing web-focused benchmarks. They also highlighted the practical value of the human-in-the-loop study and the novelty of the Non-Destructive Editing Consistency (NDEC) metric. Initial concerns were raised regarding task ambiguity, environment consistency, and the robustness of results across different UI layouts or Photoshop versions.

Following the rebuttal, reviewers (rzWc, oiMi, mHAu) noted that the authors successfully addressed key concerns through additional experiments and technical clarifications. Specifically, the authors demonstrated that PSBench is robust to UI layout shifts and software version updates (Photoshop 2020), with success rate fluctuations within 3%. They clarified that their environment reset mechanism involves restarting Photoshop after each task to ensure independence. To address scalability, the authors introduced an automated LLM-as-a-judge evaluation protocol that showed strong agreement with human annotations. While Reviewer rzWc noted that some detailed questions regarding pixel-level SSIM checks remained partially addressed, all reviewers maintained positive recommendations, acknowledging the valid core contribution and the dataset's utility for the community.

The AC recommends acceptance based on the potential of the PSBench benchmark in advancing GUI agent evaluation for complex professional applications, while also taking the constructive author-reviewer discussion into account.